# GSM-∞: How Do your LLMs Behave over Infinitely Increasing Reasoning Complexity and Context Length?

Yang Zhou [1]   Hongyi Liu [1]   Zhuoming Chen [1]   Yuandong Tian [2]   Beidi Chen [1]

## Abstract

Long-context large language models (LLMs) have recently shown strong performance in information retrieval and long-document QA. However, to tackle the most challenging intellectual problems, LLMs must reason effectively in long and complex contexts (e.g., frontier mathematical research). Studying how LLMs handle increasing reasoning complexity and context length is essential, yet existing benchmarks lack a solid basis for quantitative evaluation. Inspired by the abstraction of GSM-8K problems as computational graphs—and the ability to introduce noise by adding unnecessary nodes and edges—we develop a grade-school math problem generator capable of producing arithmetic problems with infinite difficulty and context length under fine-grained control. Using our newly synthesized GSM-∞ benchmark, we comprehensively evaluate existing LLMs. We find a consistent sigmoid decline in reasoning performance as problem complexity increases, along with a systematic inference scaling trend: exponentially increasing inference computation yields only linear performance gains. These findings underscore the fundamental limitations of current long-context LLMs and the key challenges in scaling reasoning capabilities. Our GSM-∞ benchmark provides a scalable and controllable testbed for systematically studying and advancing LLM reasoning in long and complex contexts. Code open-sources at https://infini-ai-lab.github.io/gsm_infinite/.

## 1. Introduction

Recently, state-of-the-art long-context LLMs (Team et al., 2024a; MiniMax et al., 2025) have achieved astonishing

[1]Carnegie Mellon University [2]Meta GenAI. Correspondence to: Beidi Chen <beidic@andrew.cmu.edu>.

*Proceedings of the 42nd International Conference on Machine Learning*, Vancouver, Canada. PMLR 267, 2025. Copyright 2025 by the author(s).

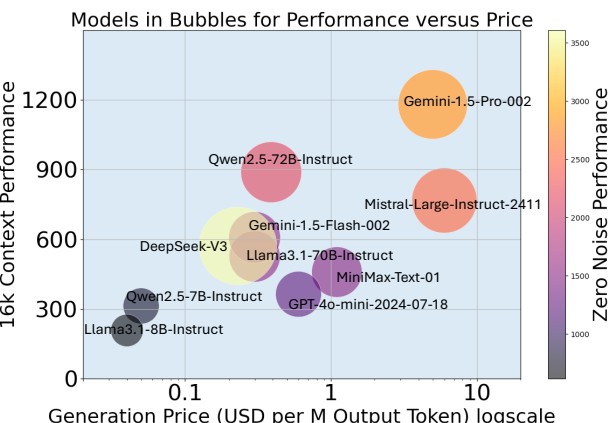

*Figure 1.* Evaluation of 10 powerful LLMs on GSM-∞, comparing API generation cost (horizontal axis) with zero-context reasoning ability (vertical axis). Bubble size represents reasoning performance at a 16K context length.

performance in a tremendously long context, where Team et al. (2024a) achieves near-perfect performance in 10M multimodal retrieval and long document QA. However, for long-context LLMs to contribute to cutting-edge mathematical and scientific discoveries or function as autonomous agents, they must be capable of processing dense, complex information and reason through multi-step tasks. For instance, Sir Andrew Wiles' proof (Wiles, 1995) of Fermat's Last Theorem in 1995 spans more than 88K highly compact tokens with deep logical connections, making context-level RAG (Lewis et al., 2021) **insufficient** and highlighting the need for long-context LLMs. Therefore, it is crucial to benchmark and facilitate long-context LLMs for complex reasoning and high-density information processing.

Although widely used, current long-context benchmarks do not fully capture the true potential of long-context LLMs (Yu et al., 2024a; Li et al., 2024a;b), making it challenging to measure their progress toward advanced intellectual agents. It is mainly due to the following three reasons: (1) **Low Complexity.** Many long-context benchmarks, such as Long-Bench (Bai et al., 2025; 2024) and most tasks in RULER (Hsieh et al., 2024b), focus on retrieval or summarization, which involve low reasoning complexity. Similar to (Yu et al., 2024b), we found that simple context-level RAG

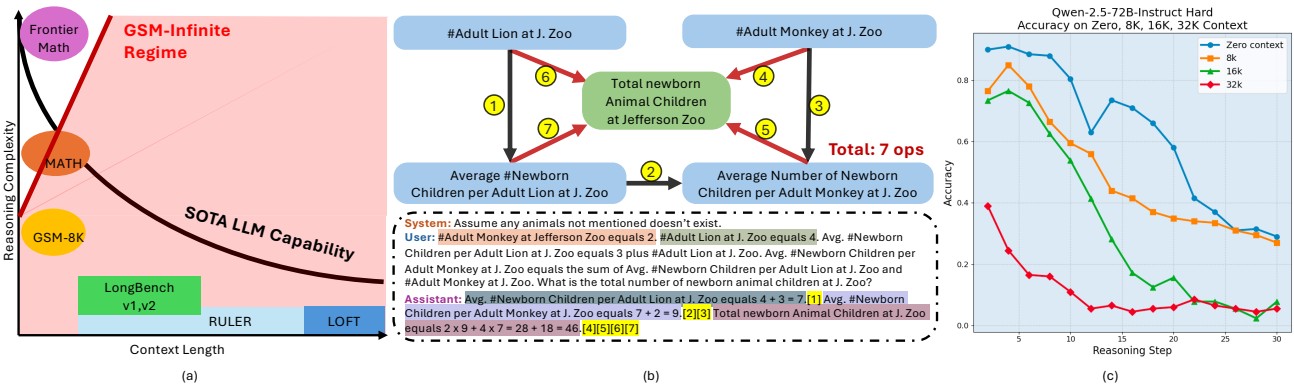

*Figure 2.* (a) We position existing benchmarks across the Reasoning complexity versus context length plot. Reasoning datasets are usually of very short context. Existing long context benchmarks are usually low in reasoning complexity. Our task can cover any context length that the user so chooses and can generate infinite reasoning complexity. However, for high reasoning complexity, our task needs to use a longer context for problems. Our task is shown in Red. (b) A simplified example of our dataset-building process. We first generate an interconnected computational graph, and we then based on the graph, attach real-world context to it to formulate the problem statements. (c) Shows Qwen2.5-72B-Instruct Score decay across zero-context, 8K, 16K, and 32K.

achieves on-par or even better results than long-context LLMs (shown in Figure 3). (2) **Detectable Noise.** Many tasks are innately short-context but are bloated into longer context through semantically irrelevant filler text (Kuratov et al. (2024) and variable-tracing in Hsieh et al. (2024b)), which is easily distinguished by a retriever of context-level RAG. (3) **Low Resource.** While complex long-context tasks exist, such as long code completion (Loughridge et al., 2024), they lack sufficient high-quality examples with adequate and verified annotation and labeling. This scarcity limits test diversity and fine-grained difficulty assessment, reducing their effectiveness in model evaluation.

Ideally, a long-context reasoning benchmark should (1) offer controllable and **scalable complexity**, (2) incorporate **hard-to-distinguish noise**, and (3) support **infinite data** generation for continuous and adaptable evaluation. Inspired by Delétang et al. (2023); Ye et al. (2024a), we model reasoning problems as computational graphs attached with language semantics. By adjusting their structure and complexity, we gain fine-grained control over reasoning difficulty and enable infinite scaling. Instead of inserting semantically irrelevant filler text, noise is introduced as additional nodes and edges upon the core graph, strategically connected to existing nodes without affecting to the necessary reasoning steps for solving the tasks. This design enables the generation of arbitrarily long test examples, difficult to differentiate relevant information from noise by context-level RAGs.

However, several technical challenges must be addressed to construct a practical benchmark. First, ensuring diverse reasoning patterns and operations - ideally providing a comprehensive coverage of GSM-8K - is crucial to enable fair and thorough evaluation. Second, computational graphs must be effectively translated into natural language that is both human- and LLM-readable, eliminating ambiguity

from memorization and ensuring a clear focus on reasoning.

We introduce GSM-∞, a long-context benchmarking framework that scales and controls reasoning complexity and noise through fine-grained manipulation of computational graphs, enabling their translation into diverse, human- and LLM-readable problems (Figure 2(b)). Specifically,

- in Section 3.1, 3.2, we discuss how to control and scale reasoning complexity by manipulating computational graphs and how to insert noise.

- Section 4.1 discusses methods to ensure comprehensive coverage of reasoning patterns appeared in GSM-8K.

- Section 4.2 shows ways enabling the computational graphs to LLM-understandable natural language mapping.

Figure 2(a) shows where our benchmark positions among existing benchmarks, demonstrating the effectiveness of GSM-∞ for long-context reasoning evaluations.

We conduct a comprehensive evaluation of 17 state-of-the-art LLMs on zero-noise problems and 10 LLMs on various noise-injected tasks using GSM-∞. We inclusively covered a wide range of models, including both popular closed-source and open-source options, conventional transformers and hybrid architectures, models of varying sizes, and both reasoning and non-reasoning LLMs. In zero-noise settings, recent reasoning-optimized LLMs demonstrate substantial improvements over their non-reasoning counterparts. Notably, Deepseek-R1 (DeepSeek-AI et al., 2025) achieves an average AUC score nearly four times higher than previous SOTA models. However, in noise-injected scenarios, LLMs exhibit varying degrees of performance degradation. Our analysis reveals several key observations:

- **Decay with reasoning complexity:** LLM performance follows a strikingly consistent sigmoid or exponential

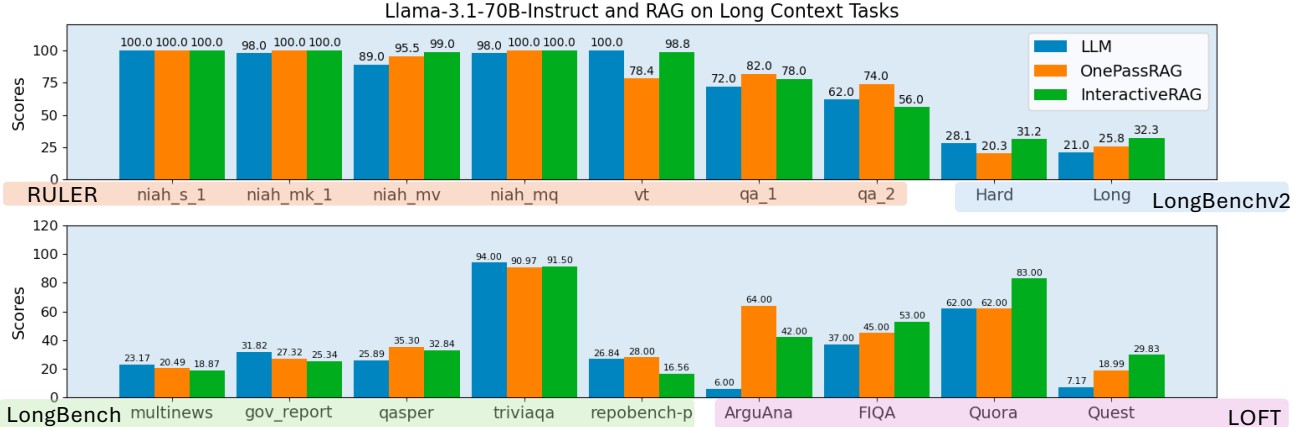

*Figure 3.* Study of Llama3.1-70B-Instruct with Passive RAG (referred to as OnePassRAG) and Active RAG (referred to as InteractiveRAG) on popular long-context benchmarks: RULER (at 64K context length), LongBench (>8K), LongBenchV2, and LOFT (128K context length). RAG is under the 2048 retrieved token budget, and the decoder used for the RAG is Llama-3.1-70B-Instruct. RAGs generally have robust performance, on par with the corresponding LLMs, showing that previous long-context benchmarks are either too simple in reasoning complexity or contain detectable noise.

decay as problem difficulty increases, highlighting fundamental limitations in scaling reasoning capabilities.

- **Decay with noise:** Performance degradation intensifies as context length increases within the same difficulty level, while longer context brings sharper degradation of the LLM performance.

- **LLM thinking patterns:** Models consistently perform better on forward-thinking tasks than on backward-thinking ones, suggesting a fundamental asymmetry in their reasoning strategies.

- **Influence of Repeated sampling:** Repeated sampling (Brown et al., 2024) on GSM-∞ reveals a clear pattern: performance improves linearly with increased inference steps but at an exponentially growing computation cost, underscoring inefficiencies in current LLM inference strategies.

## 2. Related Work and Problem Statement

Despite the wide popularity of some existing long-context benchmarks, this section reveals and elaborates on the three key limitations: low complexity, detectable noise, and low resource or limited quantity of test examples. These three limitations make it extremely challenging to measure long-context LLMs' progress toward advanced intellectual agents using existing benchmarks.

**Low Complexity.** A significant portion of long-context evaluation datasets, including RULER (Hsieh et al., 2024b), LongBench (Bai et al., 2024), LongBench v2 (Bai et al., 2025), and LOFT (Lee et al., 2024), primarily assess retrieval and summarization rather than complex reasoning.

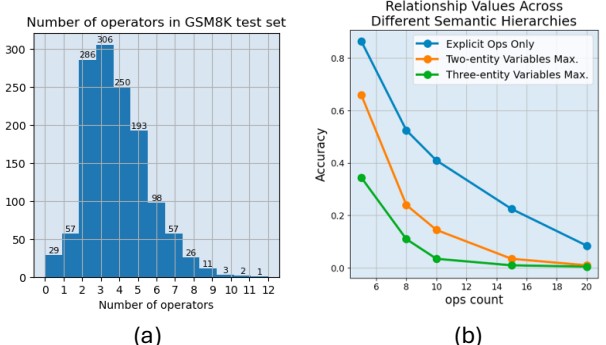

*Figure 4.* (a) presents a conservative estimate for each problem difficulty in GSM-8K 1.3K test set. We evaluate the difficulty of the problems by the number of operations needed to get to the final answer. The op count ranges from 2 to 12, while most are around 3-4. (b) shows the Llama3.1-8B-Instruct performance across different semantics hierarchies, revealing the hidden reasoning difficulty innate in natural language.

As shown in Figure 3, our experiments demonstrate that RAG systems achieve competitive results with Llama3.1-70B-Instruct across these datasets. Notably, RAG outperforms LLMs in retrieval-focused tasks (e.g., RULER, LOFT) and performs comparably in text summarization and QA (most LongBench tasks), as well as structured reasoning problems such as variable tracking (RULER-vt) and code completion (LongBench-repobench-p). RAG provide a strong baseline while being substantially more efficient.

**Detectable Noise.** Noise consists of irrelevant details that divert the model from the core task. **Many long-context**

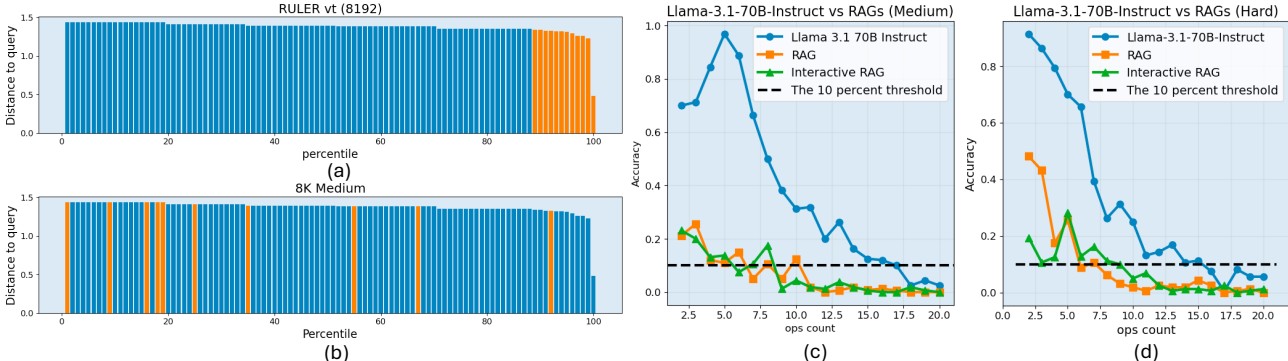

*Figure 5.* RAG performance on our proposed long-context benchmarks. (a) studies retriever's behavior on the first 100 chunks of a random problem in vt from RULER with 8192 context length. The chunks that need to be retrieved to solve the problem are labeled in coral, while the noise is in blue. The chunks have retriever scores ranked from large (semantically far) to small (semantically close). Retriever locates the essential chunks with high precision, classifying all necessary chunks with the right side of the spectrum; (b) contrasts vt with our long-context benchmarks, showing that the retriever cannot locate precisely which chunk to retrieve. (c) and (d) display the performance of two RAG systems on our benchmark medium and hard tasks. (Figure best viewed in color)

**benchmarks artificially extend short-context tasks by injecting extraneous filler text (referred as noise or distractor) that does not contribute to solving the problem, allowing retrieval-based models to filter out noise effectively.** In RULER's variable-tracing task with an 8192-token context, Llama3.1-70B-Instruct achieves 100%, while OnePassRAG and InteractiveRAG reach 82.4% and 98.4%, respectively, despite using only a 2048-token retrieval budget. A detailed breakdown in Figure 5 (a) reveals that retrievers consistently identify and prioritize relevant information while disregarding injected noise. These findings indicate that existing long-context benchmarks do not adequately justify the need for expensive long-context LLMs, as RAG systems effectively mitigate the noise impact and achieve similar performance.

**Low Resource.** Many high-quality reasoning tasks (Math and coding) heavily rely on human annotation and labeling and have test examples in limited quantity. Here we use GSM-8K (Cobbe et al., 2021a) as an example, shown in Figure 4(a). It is infeasible to extract subsets of examples with exact op at 8 for precise LLM evaluation due to the limited number of available cases—only 26 in total, with even fewer satisfying op ≥ 8. This scarcity makes meaningful evaluation impractical. Also, DafnyBench (Loughridge et al., 2024), a high-quality long-context coding benchmark only contains 782 verified and deduped examples.

**Problem Statement -** *How can we develop a benchmark that contains sufficient problems at every fine-grained level of reasoning difficulty, from easy retrieval tasks to infinitely hard challenges, while providing infinitely customizable context length with high information density?*

# 3. Computational Graphs

In this section, we detailed the construction of computational graphs, the key construct that enables GSM-∞ generation of infinite quantities of arbitrary context length and reasoning complexity. Specifically, we explain in detail the potential mapping between reasoning problems and computational graphs in Section 3.1, and in Section 3.2, we propose to generate indistinguishable noise by strategically extending the computational graph.

## 3.1. Graph Construction to Build Reasoning Problems

After carefully studying GSM-8K problems, we draw the following crucial observations, which allow us to map a randomly generated computational graph to grade-school-level math reasoning problems that cover all possible operations and relationship types.

**Mapping Explicit Ops to Computational Graphs -** From Every operation used in the GSM-8K is one of the four "+", "−", "×", and "÷". Consider the following example when operations are presented explicitly, "Eggs cost twice as much as tomatoes, while tomatoes cost 1 dollar each." These statements mention operations ("plus", "more", "times", etc.) can easily be abstracted out as a computational graph with variables, "dollar per egg" and "dollar per tomato", as nodes. There are two edges one pointing from "dollar per tomato" to "dollar per egg", while another one from a constant 2 to "dollar per tomato". Another similar example is shown in Figure 6 (a). Therefore, randomly generating a computational graph with different topology of edge connections will lead to a new reasoning problem once the natural language context are attached to the nodes of the graph.

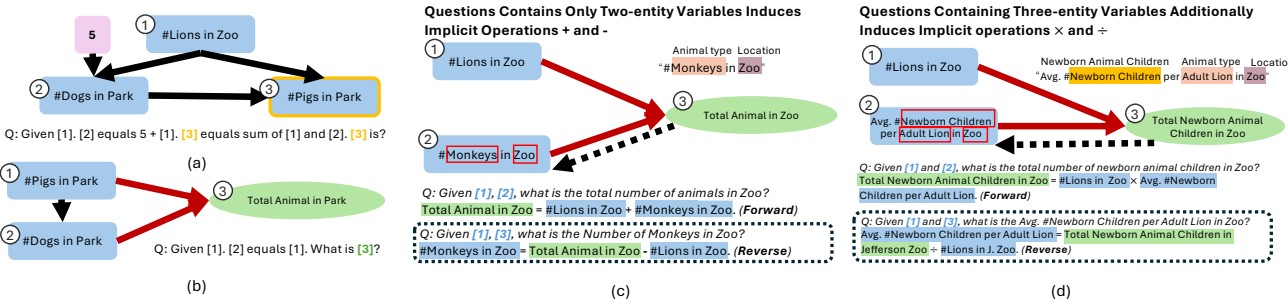

*Figure 6.* Computational Graphs Illustration. (a) shows a simple computational graph, where every node and edge can be straightforwardly converted to a statement, the other way conversion works as well. (b) breaks down the essence of implicit operations and the abstract parameter constructions; here, an assumption provided in the system prompt is to assume the animals not mentioned in question don't exist; essentially, problems in natural language omit the two red arrows in the computational graph. (c) provides an example problem when all variables that appeared are "two-entity" variables, where only implicit addition/subtraction can be generated. (d) contrasts (c) and shows an example that with additional "three-entity" variables, the computation graph can also generate implicit multiplication/division. Both (c) and (d) also illustrate the design of reverse mode that specifically aims to generate implicit subtraction and division. (Figure best viewed in color)

**Generating Implicit + using Computational Graphs -** On the other hand, the operations can also be presented implicitly hidden in natural language hierarchies. "Mary earns 20 dollars in the morning, while she earns 25 dollars in the afternoon. How much total she earned that day?" Although the problem doesn't explicitly mention addition, the solution has to sum up 20 and 25 to get 45. The reason is that natural language assumes a working day consists of morning and afternoon. Similarly, all four operations can be hidden in natural language hierarchies. Inspired by Ye et al. (2024a), we adopt its construct of "Abstract Parameters" and "Instance Parameters" to construct computational graphs that facilitate the generation of problem statements containing the hidden operations. Essentially, the newly added constructs can be thought of as adding the "total money" as a new node to the computational graph, which has two edges coming in, one from node "Morning money" and the other one from node "Afternoon money". But when generating the problem, we omit the description of two edges pointing to the node "total money on Friday". We also illustrate it in Figure 6(b). Following (Ye et al., 2024a), we connect the specific instance parameters to the abstract parameters using red edges. To reiterate, these edges are omitted, inducing implicit operations solvable only by LLM commonsense.

**Generating Implicit × using Computational Graphs -** every variables in the above-mentioned problem with the implicit "+" operation are "two-entity variables". "Morning Money" contains two entities, "Morning" and "Money", where in the context, "Money" is an attribute of "Morning". Same with "After- noon Money". In fact, out of all the examples we manually examined in GSM-8K, the minimum number of entities in the variable name is two. However, if every variable in the problem are two-entity" variables, only

generate implicit operations of + and - will be induced, but not × and ÷. For a problem to contain implicit operations ×, problems must contain variables with more than two entities in its name. For example, "Mary works 8 hours on Friday. Her hourly rate on Friday is 10 dollars. How much she will earn on Friday in total?" The variable "money per hour" contains "money", "hour", and "Friday" three entities, where "money" is an attribute of "hour", while "hour" is also an attribute of "Friday". We further contrast the above scenario in Figure 6(c) and (d), we see that if every variable is two-entity or "Animal" in "Location", generating × isn't straightforward, but once we add in the third entity, or an attribute of animal, generation of × becomes natural.

**Scaling up Reasoning Complexity and maintaining control with Computational Graph -** Our computational graph generator employs the abstract parameter construct to generate implicit operations and three-entity variables to represent multiplication operations. During the generation process of a problem, specifically, a query node is first sampled from the graph. The corresponding topological sort list—ending with the query—ensures the shortest solution path, serving as a measure of the problem's reasoning complexity. This approach enables the generation of a vast number of synthetic graphs. Furthermore, adjusting the number of variables provides a coarse control over complexity, which is refined through precise filtering to produce well-defined subsets of graphs that meet specific operation constraints.

### 3.2. Noise Construction Using Computational Graphs

**Spider Topology -** We observe that we can view noise as extending the computational graph to incorporate fake and unnecessary parameters and operators. However, two

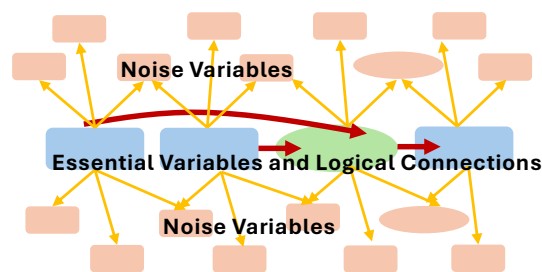

*Figure 7.* The Illustration of Noise as Extension of Core Computational Graph.

critical questions emerge. First, how to extend the computational graph without contaminating the original graph's solution and contaminations? We found that edges have to point outwards from the nodes in the original graph to the newly added noise nodes, essentially preventing the noise nodes from contributing to the core graph. Second, how to maximize the chance that RAG cannot retrieve the essential graph? It turns out interconnecting edges between newly added noise nodes won't contribute help detering RAG's retriever. We find out a simple trick works well: ensuring the majority of the added edges connect core nodes and the noise nodes contribute to a semantically close noise. We call this design Spider Topology as shown in Figure 6(c).

We evaluated the resulting noise using two RAG systems presented in 2. The results are shown in Figures 5(c) and (d). Llama3.1-70B-Instruct achieves drastically stronger performance than the two RAG systems on 8K 2-entity problems and 3-entity problems. We also carried out the same study before on our data set setting 2, in Figure 5(b). **We found that the RAG retriever now completely cannot distinguish which essential chunks from noise chunks, showing a clear contrast with vt tasks of the same context length in (a).**

## 4. GSM-∞

In this section, we present key techniques that enable the synthetic dataset to be diverse in operations in Section 4.1, LLM-understandable, and enable the evaluation to be free from non-reasoning factors in Section 4.2. Then, we present synthetic problem generators capable of generating grade-school math questions with arbitrary reasoning difficulty and context length. Thus, we generate a suite of benchmarks called GSM-∞ detailed in Section 4.3.

### 4.1. Challenge 1: How to Generate Implicit − and ÷ Operations?

**Firstly, we review why the abstract-instance construct can only generate "+" and × but not "−" and ÷.** The key limitation of Ye et al. (2024a) abstract parameters and

instance parameter design is that it is only able to generate problems with solutions with the "forward" and constructive ordering. Shown in Figure 6 (a) and (b), the design dictates that the specific and detailed variables should be defined before a more abstract variable. For example, "the number of Lions in Zoo" and "the number of Monkeys in Zoo" have to be defined before "Total Animal in Zoo" is defined. The "forward" ordering leads to the inability to generate implicit '−' operations for 2-entity problems and implicit "÷" operations for 3-entity problems that require the more abstract variables, e.g. "Total Animal in Zoo", to be defined before a more specific variable, e.g. "the number of Monkeys in Zoo".

**To generate all four kinds of implicit operations, we introduce a "reverse mode" to generate the computation graph.** Essentially, the graph construction still continues as before: starting with specific detailed variables and growing to incorporate more abstract variables. When it completes and we know all the values of nodes in the graph, we then randomly mask out specific initial low-level variables and force the solution to traverse in the reverse direction as in the "forward" ordering. We present the illustration of data generation in Figures 6 (a) and (b) for the 2-entity and 3-entity, respectively.

### 4.2. Challenge 2: How to Ensure LLM-understandable Problem Generation?

Mapping computation graphs to natural language is critical for evaluating LLMs' reasoning capabilities. **To automate this process, we develop inter-swappable templates that enhance linguistic diversity while maintaining clarity.** Several key considerations inform our design.

First, Certain syntactic forms, such as possessive constructions (e.g., A's B), are straightforward to encode but can mislead LLMs due to their deviation from natural language. For example, South Zoo's Penguin is restructured as Penguin in South Zoo, and South Zoo's Adult Penguin's Average Number of Newborn Children becomes Average Number of Newborn Children per Adult Penguin in South Zoo. Through extended trial and error, variable names with different entity numbers are **presented naturally** in all prepared templates.

Second, the number of constraints applied to the random graph generation process is as little as possible, forcing templates to **enforce unit consistency** across two-entity and three-entity variables to enable assignment between these two. For instance, "The average number of animal children per penguin in South Zoo" must share a unit with "The number of penguins in South Zoo" to allow variable assignments.

Third, to ensure real-world knowledge doesn't confuse the LLM's decision, **we avoid specific real-world locations,**

**people's names, and festival names** from appearing in the template. Based on these constricts, we propose three different templates that meet real-world templates: children-animal-zoo, teachers-school-district, and awards-movies-festival. We present in Appendix G an ablation study showing that three templates are consistent in overall performance with only minor fluctuations when evaluated using Llama-3.1-8B-Instruct. At each op, equal problems are tested for both constructive ordering (forward) and reverse ordering (reverse).

### 4.3. Benchmark Details

With the synthetic problem generators detailed in Section 4, we then use them to generate problems to build a suite of reasoning tasks with increasing complexity. For the brevity of reference, we refer to the generated problems with only explicit operations as "Easy", the generated problems with 2-entity variables at maximum as "Medium", and the generated problems with 3-entity variables at maximum as "Hard.

Ideally, when evaluating an LLM, we want to evaluate all difficulty levels, from the most basic logic complexity to when it completely fails to solve any problem. For the Easy subset of problems, it usually leads to large operation counts for powerful LLMs. However, although complexity-wise not challenging, LLMs trained with internal COT tend to generate very long arguments, saturating their API output generation limit (4K for many models). Thus, we observe a sudden decay in accuracy in large ops, not because of LLMs' ability bottlenecks, but because of the above-mentioned nuance. Thus, we make a tweak to its problem: Instead of asking the LLM to find the value of one variable, we ask the LLM to find all the variables that have some value specified, effectively increasing the difficulty of the problem.

For the modified Easy subset, we keep the generated problem in the most basic form: symbolic assignment. The typical problem statement then becomes "v1235 equals v1468 plus 1." Since the modified problem is not easier compared to Medium and Hard, we now call it "Symbolic". For Medium and Hard, we used all three templates and mixed the generated problems together to ensure diversity. For reporting LLM performance, we use **Area Under Curve (AUC)**, as shown in Figure 10, which is computing a Riemann sum over the LLM's performance in accuracy versus the number of operations from 2 to when its performance is lower than 5%.

We prepare zero-noise, 8K, 16K, and 32K in the benchmarks. The existing generation pipeline is capable of generating in > 16M context, but the smaller 70B level models effectively failed in the 32K context already, while evaluating larger ones brings cost beyond our acceptance.

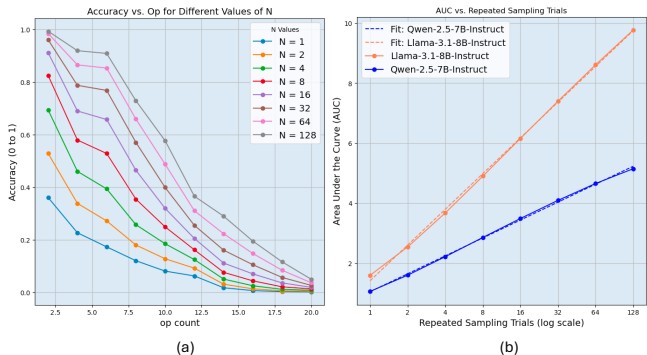

*Figure 8.* (a) shows repeated sampling on zero-context Hard task with Qwen-2.5-7B-Instruct; (b) shows the AUC to repeated sampling number of trials. We show that for repeated sampling, exponentially increasing inference compute only leads to a linear increase in AUC improvement.

## 5. Evaluation

In this section, we present comprehensive evaluations of various LLMs on GSM-∞. Specifically, the section is organized as follows:

- Section 5.1 presents the complete comparison of LLMs on both zero-noise tasks and long-context tasks. Besides the main leaderboard, we further share four interesting findings.
- Section 5.2 reveals that the sigmoid function generally fits the LLM performance degradation to the increasing ops well.
- Section 5.3 shows that LLMs generally and consistently perform forward problems better than reverse problems. (Defined in 4.1)
- Section 5.4 discuss that various LLM performance decay over longer context and further ablation of the noise.
- Section 5.5 shows that on GSM-∞, exponential increase in inference compute yields linear AUC gains.

### 5.1. Leaderboard

We evaluated 18 powerful LLMs on zero-noise problems, resulting in Table 1, while 10 models are evaluated on the long context as shown in Table 2. From Table 1, we can see that the score separates these LLMs into clear groups. Reasoning models (R1 and o1-mini) are significantly ahead of the rest of non-reasoning LLMs. On the other hand, models with hybrid architecture aren't performing strong on the zero-noise pure reasoning benchmarks. MiniMax-Text-01 and Jamba both severely underperform compared to 70 B-level models. Also, similar things can be discussed when comparing 70B and 7B level models. In Table 2, we see that the models show a very different decay pattern, while Gemini-1.5-Pro is significantly ahead of the rest of the models. Reasoning model evaluation remains too costly

*Table 1.* 18 selected models are evaluated on GSM-∞ zero-noise benchmarks using Area-Under-Curve (AUC), which is computed by taking the Riemann Sum of accuracy versus op count from 2 to when the model accuracy drops below 5%. We also present detailed statics of the first op number for the model to have an accuracy lower than 50%, 10%, and the average accuracy of the first 30 ops settings. Besides, we also highlight the reasoning models, linear attention hybrid models, and SSM hybrid models. Due to space constraint, "Mistral-Large-Instruct-2411" is shortened as "Mistral-Large";"Claude-3.5-Sonnet" and "Claude-3.5-Haiku" has version number 20241022; "GPT-4o-2024-11-20" is shortened as "GPT-4o" and "GPT-4o-mini-2024-07-18" is shortened as "GPT-4o-mini".

| Models | Three Subtasks | | | Detailed Statistics on Hard Subtask | | | Score |
| --- | --- | --- | --- | --- | --- | --- | --- |
| | Symbolic | Medium | Hard | 1st<50% op | 1st<10% op | Avg. Acc op≤30 | Avg.↑ |
| DeepSeek-R1 | 7280.0 | 9750.85 | 8573.8 | 100 | >130 | 0.9427 | 8534.88 |
| GPT-o3-mini | 6690.0 | 8335.66 | 5769.96 | 70 | 110 | 0.9423 | 6931.88 |
| GPT-o1-mini | 5060.0 | 6054.91 | 3738.43 | 50 | 90 | 0.8397 | 4951.11 |
| DeepSeek-V3 | 4310.0 | 4100.81 | 2407.86 | 24 | 55 | 0.6669 | 3606.22 |
| QwQ-32B-preview | 3530.0 | 3205.75 | 1846.19 | 21 | 50 | 0.5403 | 2860.65 |
| Gemini-1.5-Pro-002 | 2547.0 | 3659.59 | 2318.28 | 26 | 45 | 0.6924 | 2841.62 |
| Claude-3.5-Sonnet | 2161.0 | 3281.8 | 2115.79 | 26 | 40 | 0.6758 | 2519.53 |
| Mistral-Large | 2332.5 | 2879.92 | 2310.49 | 24 | 50 | 0.6645 | 2507.64 |
| Qwen2.5-72B-Instruct | 2048.0 | 2496.81 | 2016.38 | 21 | 40 | 0.5433 | 2187.06 |
| GPT-4o | 2379.0 | 2457.37 | 1451.54 | 18 | 30 | 0.5064 | 2095.97 |
| Gemini-1.5-Flash-002 | 1970.0 | 1478.75 | 1274.25 | 13 | 30 | 0.4460 | 1574.33 |
| Llama3.1-70B-Instruct | 1769.0 | 1650.25 | 1205.25 | 15 | 30 | 0.4314 | 1541.50 |
| MiniMax-Text-01 | 1618.5 | 1712.64 | 1178.51 | 14 | 30 | 0.4213 | 1503.22 |
| GPT-4o-mini | 1389.0 | 1406.5 | 913.89 | 12 | 22 | 0.3094 | 1236.46 |
| Claude-3.5-Haiku | 897.0 | 1053.16 | 784.34 | 10 | 22 | 0.2910 | 911.50 |
| Qwen2.5-7B-Instruct | 786.95 | 886.75 | 618.5 | 7 | 19 | 0.2257 | 764.07 |
| Llama3.1-8B-Instruct | 462.0 | 786.5 | 606.5 | 6 | 17 | 0.2186 | 618.30 |
| Jamba-1.5-Large | 856.0 | 485.13 | 466.4 | 6 | 26 | 0.1828 | 602.51 |

*Table 2.* 10 selected models are evaluated on GSM-∞ Long Context benchmarks using Average AUC of Symbolic, Medium, and Hard. We evaluated models on 8K, 16K, and 32K context. Although our pipeline is capable of generating longer problems, the resource required to go further for larger models beyond our acceptance, while smaller models effectively has completely failed.

| Model | 8K | 16K | 32K | Avg.↑ |
| --- | --- | --- | --- | --- |
| Gemini-1.5-Pro-002 | 1182.43 | 896.31 | 812.96 | 963.9 |
| Qwen2.5-72B-Instruct | 927.33 | 681.53 | 563.65 | 724.17 |
| Mistral-Large | 914.49 | 563.73 | 319.21 | 599.14 |
| DeepSeek-V3 | 935.10 | 477.02 | 313.66 | 575.2 |
| Gemini-1.5-Flash-002 | 673.88 | 476.72 | 377.38 | 509.3 |
| Llama3.1-70B-Instruct | 479.00 | 394.50 | 355.5 | 409.67 |
| MiniMax-Text-01 | 481.32 | 359.56 | 325.95 | 388.94 |
| GPT-4o-mini | 401.00 | 337.81 | 275.63 | 338.15 |
| Qwen2.5-7B-Instruct | 248.00 | 211.50 | 196.17 | 218.56 |
| Llama3.1-8B-Instruct | 183.67 | 149.50 | 109.45 | 147.54 |

or slow for current long-context leaderboard participation.

Later on, we examine the behavior of LLMs on GSM-∞ and summarize four interesting findings uniquely enabled by our method of problem generation.

## 5.2. LLM Performance Degradation Can be Modeled Using Sigmoid Function

Our construction of GSM-∞ enables precise measurement of LLM performance across fine-grained difficulty levels. For each subtask, we observe a clear trend: LLM accuracy declines as the number of required operations increases. Sur-

prisingly, most models exhibit a sigmoid-like performance decay, as shown in 9 for forward problems (see Section 4.1 for definitions).

Under the Medium subtask, LLM performance aligns remarkably well with a sigmoid function curve, with $R^2 > 0.98$. The pattern is intuitive: at low operation counts, accuracy remains near 1.0; as complexity increases, performance first decays gradually, then drops sharply toward zero, where LLMs effectively fail to solve the problems. The score eventually stabilizes near zero. Interestingly, while LLMs vary in overall capability, they follow the same trend, differing primarily in the decay rate of the sigmoid function.

## 5.3. Reverse Problems are Harder to Solve for LLMs

The generator of GSM-∞ can generate both the "forward" and the "reverse" problems, which can be compared separately. Most LLMs perform worse in reverse problems than forward ones, shown in Figure 9(b) using Mistral-Large (Jiang et al., 2023a). For the Medium, only Jamba-1.5-Large out of a total of 18 LLMs evaluated do reverse problems better than forward problems, and the average difference in AUC is 604. For the Hard subtask, 5 out of 18 models have larger reverse problems AUC, Deepseek R1, Gemini-1.5-Pro-002, Qwen2.5-7B-Instruct (Qwen et al., 2025), 4o-mini, and Jamba-1.5-Large (Team et al., 2024b). The average difference in AUC is 154. A detailed breakdown is listed in Appendix F. Besides, LLM performance

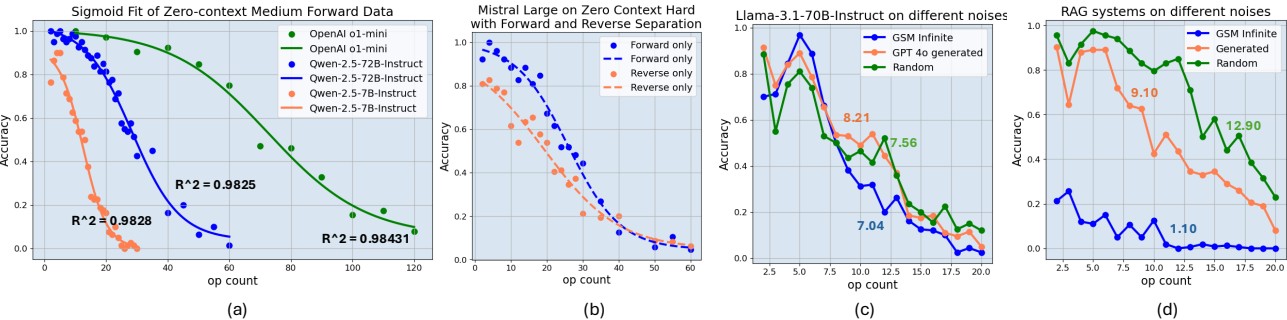

*Figure 9.* (a) shows three different LLMs' behavior on GSM-∞ benchmark zero-context Medium Forward, GPT o1-mini, Qwen2.5-72B-Instruct, and Qwen2.5-7B-Instruct. These models are drastically different in reasoning ability but have performance be modeled by sigmoid well, all with $R^2$ >0.98. (b) shows the gap between forward-thinking and reverse-thinking problems from Mistral Large on zero-context Hard. reverse-thinking problems are significantly harder and can be approximated by the sigmoid function that is essentially left-ward shifting from the forward sigmoid function. (c) and (d) presents RAG and corresponding LLMs' performance on different noises. Other than ours, RAG even improves performance.

on reverse problems can also be modeled by a sigmoid mapping. More plots are presented in Appendix F.

### 5.4. Long-context Degradation and Noise Ablation

We evaluate LLM performance across increasing context lengths (0, 8K, 16K, 32K) and observe a consistent decline in performance as context length increases. Notably, models exhibit different decay patterns. We present results for 10 models across 3 subtasks, each with four curves representing different context lengths. All 30 plots are in Appendix L.

We conduct an ablation study on three noise types: GSM-∞ (ours), LLM-generated, and random. For LLM-generated noise, we prompt GPT-4o to create a fake documentary-style commentary on random problems, occasionally introducing nonsensical variable mentions. For random noise, we follow Hsieh et al. (2024), using generic statements like "The sky is blue. The tree is green." We evaluate Llama3.1-70B-Instruct and a RAG system under all three noise types in an 8K context. Interestingly, RAG outperforms long-context LLMs on LLM-generated and random noise, effectively filtering irrelevant content. However, it fails to distinguish GSM-∞ noise from crucial sentences.

### 5.5. Limitations of Repeated Sampling

The construction of GSM-∞ allows us to study techniques for Inference Scaling as well. Specifically, we study repeated sampling (Brown et al., 2024; Snell et al., 2024) and its effectiveness under different reasoning complexity. We study both Qwen2.5-7B-Instruct and Llama3.1-8B-Instruct with the best-of-N settings similar to Brown et al. (2024). Interestingly, we find that repeated sampling seems to boost the performance the most for smaller op count subsets, and the benefit of repeated sampling diminishes gradually for larger op count subsets, Qwen2.5-7B-Instruct behavior is plotted in Figure 8(a) with different repeated trial settings.

Surprisingly, suppose we calculate the AUC score under every curve corresponding to each number of repeated trial settings. In that case, we find that the increment in the AUC score from two consecutive settings is close. If we plot the AUC score versus the number of repeated trial settings and take the log scale of the repeated trial N, the graph is linear, as shown in Figure 8(b). The R-squared is greater than 0.99 for both Qwen2.5-72B-Instruct and Llama3.1-8B-Instruct, where Llama3.1-8B-Instruct. **Therefore, GSM-∞ helps reveal that Repeated Sampling gives linear AUC Improvement from exponentially increasing compute cost.**

## 6. Conclusion

Long-context LLMs have the potential to tackle complex, information-dense tasks requiring deep reasoning and coherent long-form generation. To advance their development and benchmarking, we introduce GSM-∞, a synthetic long-context reasoning benchmark generated entirely by a software-based system with fine-grained control over complexity and information density. Through extensive evaluations on GSM-∞, we uncover key insights to inform future LLM training and inference improvements.

## Impact Statement

This paper presents work aimed at advancing the field of Machine Learning, specifically in long-context reasoning for large language models. Our benchmark, GSM∞, provides a scalable and controllable testbed for evaluating and improving LLM reasoning capabilities in complex, extended contexts. While this research contributes to the broader development of AI reasoning, we do not foresee any immediate ethical concerns or societal implications beyond standard considerations in AI research, none of which we feel must be specifically highlighted here.

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

## Appendix Table of Contents

We present the following table of contents to better traverse the Appendix.

## A. Related Work

### A.1. Long-context Language Models

Various works related to the Long-context Language Model have been proposed. Flash attention(Dao et al., 2022), Flash attention2(Dao, 2024), Ring attention(Liu et al., 2023a), and Tree attention(Shyam et al., 2024) significantly reduced the memory footprint and communication overhead for processing long context in engineering level across multiple nodes. Architectural level innovations such as sparse attentions represented by sliding window attention(Beltagy et al., 2020), are also widely used to reduce the overhead caused by the increasing sequence length. New training strategies, such as gradually extending the training context length in the final stages of pretraining have been applied to support a long context window(Dubey et al., 2024).

### A.2. Long context benchmarks and tasks

There have been a quite a few works benchmarking long-context language models. Existing comprehensive benchmarks like ∞bench(Zhang et al., 2024) cover realistic tasks including document QA, summary, and synthetic tasks including information retrieval, expression calculation, extending the context length in the benchmark to over 200k tokens. ∞bench(Zhang et al., 2024) does have mathematical reasoning tasks, however the most relevant math.calc part seems to be too difficult for SOTA models to work out. Synthetic tasks often offer more control and are less affected by parametric knowledge in comparison with realistic tasks. One comprehensive synthetic benchmark is RULER(Hsieh et al., 2024a), a synthetic benchmark with

tasks including retrieval, variable tracking and so on, offering some controls over context length and task complexity. Experiments with various complexities were done, but it does not provide a quantitative analysis of complexity and context length on the correctness of the task, let alone isolate two separate patterns of performance decay. Other benchmarks usually focus on simple retrieval(Github, 2023; Liu et al., 2023b), fact reasoning(Kuratov et al., 2024), the impact of long context on natural language reasoning(Levy et al., 2024) and other real-world knowledge involved tasks.

### A.3. Limitation of Existing Reasoning Tasks

Popular reasoning benchmarks are loose collections of human-made problems that naturally suffer from the following limitations. Firstly, the difficulty of problems within the same benchmark varies widely. We analyzed all 1.3K test problems in the GSM8K dataset, we plot the histogram in the number of operations in Figure 4(a). The problem varied from 2 to over 12 following a skewed bell shape curve. This lack of fine-grained control makes it challenging to systematically evaluate models across incremental difficulty levels. Also, notice that the total number of problems is less than 10 for op $\geq 9$, too little for stable evaluation. The lack of problem quantity on human-curated datasets eliminates the possibility of filtering out problems of each fine-grained difficulty level. Secondly, there is a significant difficulty gap between the benchmarks: GSM-8K focuses on middle school problems, MATH (Hendrycks et al., 2021) and AIME targets prospective university students, and Frontier Math challenges top-tier math graduate students. It is difficult to quantitatively determine the difference in problem difficulty between GSM-8K problems with MATH problems since MATH uses operations such as taking power or roots that are absent in GSM-8K. Besides, there exists a rich set of common-sense reasoning benchmarks(Yang et al., 2018; Trivedi et al., 2022; Geva et al., 2021). However, the number of hops are usually limited to two to four, which although natural and requires LLM commonsense, is significantly limited in complexity. Similarly, it is not possible to determine the difference in complexity from MATH to Frontier Math. It is difficult to quantitatively model LLMs' performance degradation with the continuously increasing difficulty of the problem. Third, most of the existing problems have very short input prompts. On average, GSM-8K test set problems have a length of 59.96 tokens, while MATH test set problems have 67.37 tokens when using Llama 3.1 tokenizer. We have seen from 2 that the addition of irrelevant noise cannot meaningfully evaluate the ability to reason in a long context of LLMs.

### A.4. Synthesized Datasets for long-context

Synthesized tasks are simple to build and absolutely deterministic, data contamination safe, but highly effective to evaluate certain aspects of LLM performance. Its use in long-context benchmarks is profound. Needle-in-the-haystack (Kamradt, 2023), a pioneering long-context task, becomes the go-to task for evaluating LLM long-context retrieval ability. On the other hand, LLM reasoning benchmarks also see recent efforts in synthesized tasks. (Sinha et al., 2019) proposes to evaluate language models through a description of kinship of many people, which can be naturally formulated in graphs. Although the problem generator proposed require human annotation for ground-truth and impose no absolute guarantee in correctness of the problems, preventing scaling up of the method. (Mirzadeh et al., 2024) recently proposes to use build synthesized dataset upon GSM8K (Cobbe et al., 2021b) to study the robustness of LLM reasoning. **Part of our work draws a strong inspiration from a series of works ((Ye et al., 2024a), (Ye et al., 2024b)) which systematically studies the intricacies of decoder transformers in solving grade-school level problems.** Following their footsteps, we carefully redesign the process of generating the problems so current LLMs can solve without training, and together with thoughtful steps in noise addition, we effectively construct effective reasoning benchmarks for the long-context community. Concurrently, (Vodrahalli et al., 2024) also introduce meaningful noise into short essential problem sentences to probe long-context ability of LLM. Differently, we also focus on scaling reasoning complexity (#hops), which help me present significantly different insights from their study into understanding both scaling up meaningful context length and reasoning complexity continuously.

## B. Benchmark Limitation

Despite our careful construction and attention to LLM understandability, there are still several important limitations to the existing benchmark GSM-∞. First, our current benchmark only targets operations contained in the grades school math, so only $+, -, \times, \div$ are included. We are aware that Math (Hendrycks et al., 2021) includes more complex operators, which we are looking at incorporating into the next version of the benchmark. The key limitation it brings to our benchmark is shown in Figure 2(a), to generate complex problems with high reasoning complexity, our generator needs to generate problems that are much longer than hard mathematic benchmarks, despite our limitless quantity and complexity scalability. Second, another key limitation of synthetic benchmarks is the lack of diversity in natural language. Natural Language reasoning tasks usually contain multiple ways to assign a variable value or to describe a relationship between multiple variables, which

is extremely difficult to program in software without the LLMs in the loop. We deliberately aim to keep LLMs out of the loop for better scalability and counter the diversity by incorporating more than one template into the problem generation. Third, we implement multiple checks to ensure all of the essential steps in the core graph have positive values and have a number range of less than four figures, but for the current noise generation, we don't implement these checks, since they are redundant to the problems. However, we do notice that when processing through the noise, the LLMs sometimes are confused by the irregular noise variable values. In later version of the benchmark, we will address this issue and make sure the entire graph is consistent.

## C. Detailed Experiment Setup

### C.1. RAG Experiment Setup

The RAG system contains two components, the retriever and the decoder. For the retriever, we use all-mpnet-v2-base. For the decoder, we use Llama-3.1-70B-Instruct. We follow the convention of context-level RAG method. We solely use the input context as the data store (no external source). We segment the input text into chunks (roughly a sentence) using NLTK package. The context retrieval budget for all problems is 2048. We employ two different RAG methods: passive and active RAGs. Passive RAG calls the retriever once before the generation of the decoder. The retriever computes the semantic similarity or distance between each chunk of context and the query sentence. These chunks in context are then ranked from closest (most semantic similar) to furthest (least semantic similar), and depending on the retrieved context, top-k chunks are retrieved. For our study, we used the L2 distance between context chunk embeddings and query embeddings. The decoder then takes the retrieved chunks as input and then outputs its response to the query.

We use two types of RAG systems: Passive RAG that only calls the retriever once at the beginning to retrieve relevant context or Interactive RAG (Jiang et al., 2023b) in which the decoder decides when to retrieve, how many retrievals are needed, and generate a query for each retrieval. For the latter one, we restrict the decoder generation with only the latest retrieval content and its past generation.

On the other hand, active RAG shows strong performance (Jiang et al., 2023b) especially for common sense reasoning tasks. In addition to the steps in passive RAGs, the decoder is allowed to initiate additional calls to the retriever to retrieve more context by generating new queries. We follow the state-of-the-art active RAG method FLARE (Jiang et al., 2023b), which allows for 10 rounds of query, but restricts the LLM to only see its current round of retrieved context and its past rounds of generation to generate its full response.

### C.2. Detailed Experiment Metrics

The Area Under Curve (AUC) metric, conceptually illustrated in Figure 10, quantifies model performance by integrating accuracy over the "Number of Ops." First, accuracy measurements ($y_i$) are collected at corresponding "Number of Ops" values ($x_i$). These ($x_i, y_i$) pairs are then sorted based on the $x_i$ values. The AUC is subsequently computed using the trapezoidal rule on these sorted points. For $N$ data points $(x_0, y_0), \ldots, (x_{N-1}, y_{N-1})$ sorted by $x$, the raw area is given by:

$$\text{AUC}_{\text{raw}} = \sum_{i=1}^{N-1} \frac{y_i + y_{i-1}}{2}(x_i - x_{i-1})$$

This sum approximates the area under the curve formed by connecting consecutive (Accuracy, Number of Ops) points. The final reported AUC value is then scaled:

$$\text{AUC} = \text{AUC}_{\text{raw}} \times 100 \tag{1}$$

A higher AUC indicates superior overall accuracy maintained across the evaluated range of "Number of Ops." The "Acc Threshold" in the figure serves as a visual performance benchmark. While Figure 10 depicts rectangular "boxes" for illustrative simplicity, the actual numerical integration employs the more precise trapezoidal method described.

### C.3. Repeated Sampling Experiment Setup

#### C.3.1. PROCEDURE

1. **Oversampling Phase**
   - Generate 256 samples per task with temperature $T = 1.0$

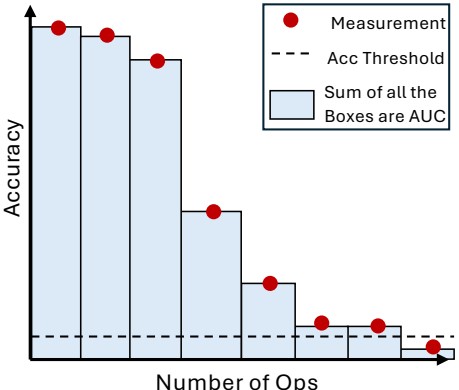

*Figure 10.* Area Under Curve Metrics is Used to Compare between LLM Performance.

- Use fixed random seeds for reproducibility

2. **Accuracy Calculation**

   - Compute per-task empirical accuracy:

$$p_{\text{task}} = \frac{\#\ \text{Correct Samples}}{256} \tag{2}$$

   - Estimate accuracy for N samples:

$$\text{Acc}_{\text{task}} = 1 - (1 - p_{\text{task}})^N \tag{3}$$

3. **Aggregation**

   - Average results across 80 tasks:

$$\text{Final Accuracy} = \frac{1}{80} \sum_{i=1}^{80} \text{Acc}_{\text{task}_i} \tag{4}$$

C.3.2. RATIONALE

- **Oversampling**: 256 samples reduces variance in estimating $p_{\text{task}}$ compared to using 128 samples directly

- **Probability Formula**: Models cumulative success probability:

$$P(\geq 1 \text{ correct in } k \text{ trials}) = 1 - (1 - p)^k$$

- **Task Count**: 80 tasks per op provide stable statistics while remaining computationally feasible

# D. Full Result of RAG experiments

Here, we present the full result of the RAG experiment for further analysis. We retrieve 2048 tokens for each RAG retrieval.

## D.1. RULER

| Models | s1 | s2 | s3 | mk1 | mk2 | mk3 | mv | mq | Context Length |
|---|---|---|---|---|---|---|---|---|---|
| Llama 3.1 70B Instruct | 100 | 100 | 100 | 100 | 100 | 100 | 100 | 100 | 8k |
| OnePass RAG | 100 | 100 | 100 | 100 | 100 | 96 | 98.5 | 100 | 8k |
| Interactive RAG | 100 | 100 | 100 | 100 | 100 | 98 | 99 | 99 | 8k |
| Llama 3.1 70B Instruct | 100 | 100 | 100 | 100 | 98 | 100 | 98 | 100 | 32k |
| OnePass RAG | 100 | 100 | 100 | 98 | 100 | 72 | 99.5 | 97.5 | 32k |
| Interactive RAG | 100 | 100 | 100 | 98 | 96 | 96 | 98.5 | 98.5 | 32k |
| Llama 3.1 70B Instruct | 100 | 100 | 100 | 98 | 96 | 100 | 89 | 98 | 64k |
| OnePass RAG | 100 | 100 | 100 | 100 | 100 | 56 | 95.5 | 100 | 64k |
| Interactive RAG | 100 | 100 | 100 | 100 | 96 | 98 | 99 | 100 | 64k |

Table 3: RAG vs Model (RULER NIAH)

| Models | vt | cwe | fwe | qa1 | qa2 | Context Length |
|---|---|---|---|---|---|---|
| Llama 3.1 70B Instruct | 100 | 100 | 96.67 | 84 | 74 | 8k |
| OnePass RAG | 82.4 | 14.8 | 97.33 | 86 | 86 | 8k |
| Interactive RAG | 98.4 | 31.2 | 79.33 | 80 | 68 | 8k |
| Llama 3.1 70B Instruct | 100 | 95.2 | 97.33 | 80 | 66 | 32k |
| OnePass RAG | 86 | 5.2 | 92 | 84 | 74 | 32k |
| Interactive RAG | 98 | 7.6 | 80 | 78 | 64 | 32k |
| Llama 3.1 70B Instruct | 100 | 6.2 | 95.33 | 72 | 62 | 64k |
| OnePass RAG | 78.4 | 1.2 | 88.67 | 82 | 74 | 64k |
| Interactive RAG | 98.8 | 2.6 | 72 | 78 | 56 | 64k |

Table 4: RAG vs Model (RULER other subsets)

**Abbreviations**: s1-3 = niah_single_1-3, mk1-3 = niah_multikey_1-3, mv = niah_multivalue, mq = niah_multiquery

## D.2. LongBench V2

| Tasks | Overall | Easy | Hard | Short | Long |
|---|---|---|---|---|---|
| Llama 3.1 70B Instruct | 30 | 33.3 | 28.1 | 44.7 | 21 |
| OnePassRAG | 25 | 33.3 | 20.3 | 23.7 | 25.8 |
| InteractiveRAG | 33 | 36.1 | 31.2 | 34.2 | 32.3 |

Table 5: RAG vs Model (LongBench V2)

## D.3. LongBench

| Tasks | passage_count | hotpot-qa | samsum |
|---|---|---|---|
| Llama 3.1 70B Instruct | 36.0,36.0,32.0 | 58.87,71.22,76.44 | 28.88,35.95,41.48 |
| OnePassRAG | 0.0,0.0,0.0 | 65.04,61.59,63.05 | 31.63,23.28,26.84 |
| InteractiveRAG | 27.0,14.0,6.0 | 61.86,51.0,55.94 | 24.83,20.21,23.81 |

Table 6: RAG vs Model (LongBench) - Part 1

| Tasks | multi-news | multifieldqa_en | gov_report |
|---|---|---|---|
| Llama 3.1 70B Instruct | 27.71,24.81,23.17 | 57.31,51.83,64.98 | 34.94,34.97,31.82 |
| OnePassRAG | 26.85,22.72,20.49 | 51.69,48.66,55.85 | 32.6,30.67,27.32 |
| InteractiveRAG | 24.64,19.99,18.87 | 47.71,42.98,58.45 | 29.91,27.02,25.34 |

Table 7: RAG vs Model (LongBench) - Part 2

| Tasks | qasper | passage_retrieval_en | 2wikimqa |
|---|---|---|---|
| Llama 3.1 70B Instruct | 50.3,46.5,25.89 | 100.0,100.0,100.0 | 74.93,64.37,59.6 |
| OnePassRAG | 45.73,43.5,35.3 | 72.0,72.0,79.0 | 67.97,59.64,48.4 |
| InteractiveRAG | 43.8,37.9,32.84 | 91.0,90.0,85.33 | 46.04,43.46,38.8 |

Table 8: RAG vs Model (LongBench) - Part 3

| Tasks | triviaqa | trec | lcc | repobench-p |
|---|---|---|---|---|
| Llama 3.1 70B Instruct | 82.0,93.6,94.0 | 48.0,12.0,12.0 | 50.14,55.0,50.04 | 29.98,27.82,26.84 |
| OnePassRAG | 92.13,89.46,90.97 | 47.0,56.0,53.0 | 19.92,14.5,18.36 | 34.76,33.62,28.0 |
| InteractiveRAG | 88.11,92.32,91.5 | 56.0,57.0,52.0 | 24.26,23.26,22.57 | 14.97,17.15,16.56 |

Table 9: RAG vs Model (LongBench) - Part 4

**Comments**: The 3 data separated by commas are subsets of 0-4k, 4-8k,8k+ respectively

## D.4. LOFT

| Tasks | ArguAna | FEVER | FIQA | MS MARCO | NQ | Quora | SciFact |
|---|---|---|---|---|---|---|---|
| Llama 3.1 70B Instruct | 0.06 | 0.78 | 0.37 | 0.67 | 0.84 | 0.62 | 0.59 |
| OnePassRAG | 0.64 | 0.88 | 0.45 | 0.77 | 0.86 | 0.62 | 0.64 |
| InteractiveRAG | 0.42 | 0.73 | 0.53 | 0.69 | 0.76 | 0.83 | 0.87 |

Table 10: RAG vs Model (LOFT) - Part 1

| Tasks | Touché-2020 | HotPotQA | MuSiQue | QAMPARI | QUEST |
|---|---|---|---|---|---|
| Llama 3.1 70B Instruct | 0.4411 | 0.37 | 0.2 | 0.024 | 0.07166 |
| OnePassRAG | 0.2529 | 0.455 | 0.2383 | 0.1559 | 0.1899 |
| InteractiveRAG | 0.79 | 0.29 | 0.13 | 0.1539 | 0.2983 |

Table 11: RAG vs Model (LOFT) - Part 2

**Comments**: Due to LOFT's limited prompt availability (no official releases for 32k/1M contexts), we conducted experiments solely on 128k context lengths. We observed that LOFT's document ID retrieval tasks inherently require document identifiers that aren't captured in standard RAG-retrieved chunks. To address this, we implemented a lightweight modification: appending document ID tags to each context chunk during retrieval. This allows the RAG system to infer the correct document ID when retrieving relevant content, resolving the task-specific limitation without altering core RAG functionality.

# E. Illustrative Problems

This section presents one representative problem from each subset (Symbolic, Medium, and Hard) defined in the appendix. These examples illustrate the variations within the benchmark. see Table 12

Table 12. Illustrative Problems from Each Subset

| Feature | Symbolic | Medium | Hard |
|---|---|---|---|
| Problem | | | |

- **Symbolic (op=5):** <context>\nassign V705804 = V437110 + 1. assign V986916 = V705804. assign V873548 = 6. assign V684196 = V873548. assign V437110 = V873548.\n </context> \n\nThe context contains relationships between variables. These relationships are independent mathematical equations that are all satisfied simultaneously.\n Using only these relationships, determine which variables (if any) from which values can be derived are equal to 7.\nShow your step-by-step reasoning and calculations, and then conclude your final answer in a sentence. **Answer**: V705804,V986916.

- **Medium (op=5):** Problem: The number of adult owl in Bundle Ranch equals 2 times the number of adult eagle in Bundle Ranch. The number of adult eagle in Hamilton Farm equals the difference between the total number of adult animals in Bundle Ranch and the number of adult eagle in Bundle Ranch. The number of adult owl in Hamilton Farm equals 4 times the number of adult owl in Bundle Ranch. The number of adult eagle in Bundle Ranch equals 3. Question: What is the total number of adult animals in Bundle Ranch? **Answer**: 9.

- **Hard (op=5):** The average number of newborn children per adult blue jay in Bundle Ranch equals 2. The number of adult parrot in Bundle Ranch equals 2. The number of adult blue jay in Bundle Ranch equals 2 times the average number of newborn children per adult blue jay in Bundle Ranch. The number of adult eagle in Bundle Ranch equals 2 times the average number of newborn children per adult blue jay in Bundle Ranch. The number of adult parrot in South Zoo equals 4 times the sum of the average number of newborn children per adult eagle in Hamilton Farm, the number of adult eagle in Hamilton Farm, and the average number of newborn children per adult eagle in Hamilton Farm. The average number of newborn children per adult eagle in Hamilton Farm equals the number of adult eagle in Bundle Ranch. The number of adult eagle in Hamilton Farm equals 3. The average number of newborn children per adult parrot in Bundle Ranch equals the total number of adult animals in Hamilton Farm. The number of adult eagle in South Zoo equals 1. The average number of newborn children per adult parrot in South Zoo equals the average number of newborn children per adult parrot in Bundle Ranch. The average number of newborn children per adult eagle in Bundle Ranch equals 3 plus the average number of newborn children per adult parrot in Bundle Ranch. The average number of newborn children per adult eagle in South Zoo equals the sum of the number of adult blue jay in Bundle Ranch, the average number of newborn children per adult blue jay in Bundle Ranch, the average number of newborn children per adult parrot in Bundle Ranch, and the number of adult parrot in Bundle Ranch. Question: What is the average number of newborn children per adult eagle in Bundle Ranch? **Answer**: 6.

# F. Forward and Reverse Problems Breakdown

| Models | Forward Problem | Reverse Problem | Forward AUC - Reverse AUC |
|---|---|---|---|
| Llama3.1-70B-Instruct | 2100.625000 | 1283.750000 | 816.875000 |
| GPT-4o-mini | 1529.725400 | 1267.579000 | 262.146400 |
| Jamba-1.5-Large | 390.380000 | 624.980000 | -234.600000 |
| GPT-4o | 3073.997375 | 1952.816875 | 1121.180500 |
| Mistral-Large | 3468.234100 | 2431.732450 | 1036.501650 |
| Llama3.1-8B-Instruct | 1030.000000 | 563.125000 | 466.875000 |
| Claude-3.5-Sonnet | 3653.830050 | 3158.657850 | 495.172200 |
| Qwen2.5-72B-Instruct | 2889.375000 | 2141.250000 | 748.125000 |
| Qwen2.5-7B-Instruct | 995.625000 | 833.125000 | 162.500000 |
| o1-mini | 6517.510550 | 5592.307100 | 925.203450 |
| Gemini-1.5-Flash-002 | 1889.375000 | 1153.750000 | 735.625000 |
| Claude-3.5-Haiku | 1234.620000 | 873.100000 | 361.520000 |
| Llama-3.1-405B-Instruct | 1781.400000 | 981.250000 | 800.150000 |
| DeepSeek-V3 | 4613.125000 | 3713.125000 | 900.000000 |
| Gemini-1.5-Pro-002 | 4204.564075 | 3160.574950 | 1043.989125 |
| DeepSeek-R1 | 9764.950000 | 9750.950000 | 14.000000 |
| MiniMax-Text-01 | 2148.071300 | 1539.415650 | 608.655650 |
| QwQ-32B-Preview | 3530.000000 | 2846.250000 | 683.750000 |

Table 13: Medium Difference in AUC in Forward Problems and Reverse Problems

| Models | Forward Problem | Reverse Problem | Forward AUC - Reverse AUC |
|---|---|---|---|
| Claude-3.5-Haiku | 819.240000 | 776.900000 | 42.340000 |
| Llama3.1-70B-Instruct | 1314.375000 | 1098.750000 | 215.625000 |
| Gemini-1.5-Flash-002 | 1341.250000 | 1219.375000 | 121.875000 |
| MiniMax-Text-01 | 1360.555000 | 1034.625000 | 325.930000 |
| DeepSeek-R1 | 8444.500000 | 8756.950000 | -312.450000 |
| o1-mini | 3831.381000 | 3645.474200 | 185.906800 |
| Gemini-1.5-Pro-002 | 2255.732025 | 2444.270375 | -188.538350 |
| DeepSeek-V3 | 2725.085000 | 2109.560000 | 615.525000 |
| Qwen2.5-7B-Instruct | 625.625000 | 630.625000 | -5.000000 |
| GPT-4o | 1592.280000 | 1311.560000 | 280.720000 |
| Llama3.1-8B-Instruct | 759.375000 | 460.625000 | 298.750000 |
| Qwen2.5-72B-Instruct | 2196.875000 | 1895.000000 | 301.875000 |
| Claude-3.5-Sonnet | 2242.309950 | 1999.998100 | 242.311850 |
| GPT-4o-mini | 858.400000 | 873.310000 | -14.910000 |
| QwQ-32B-Preview | 1878.750000 | 1855.625000 | 23.125000 |
| Mistral-Large | 2570.940500 | 2018.469000 | 552.471500 |
| Llama-3.1-405B-Instruct | 1215.000000 | 743.750000 | 471.250000 |
| Jamba-1.5-Large | 274.980000 | 699.990000 | -425.010000 |

Table 14: Hard Difference in AUC in Forward Problems and Reverse Problems

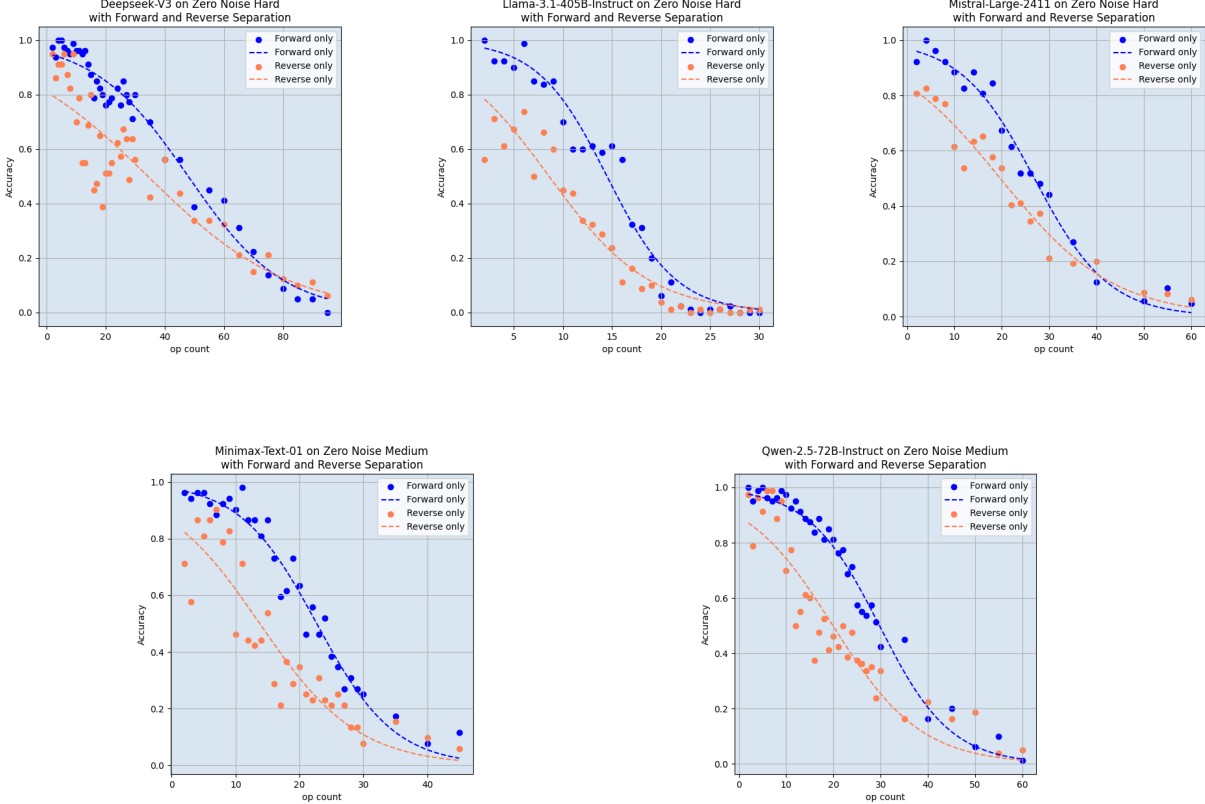

*Figure 11.* Comparison between forward and reverse

## G. Ablation Study of Task Templates

We have three different real-world templates ready. We show that they offer consistent scores with Llama-3.1-8B-Instruct, with slight variables in specific operations. We also show three problem examples.

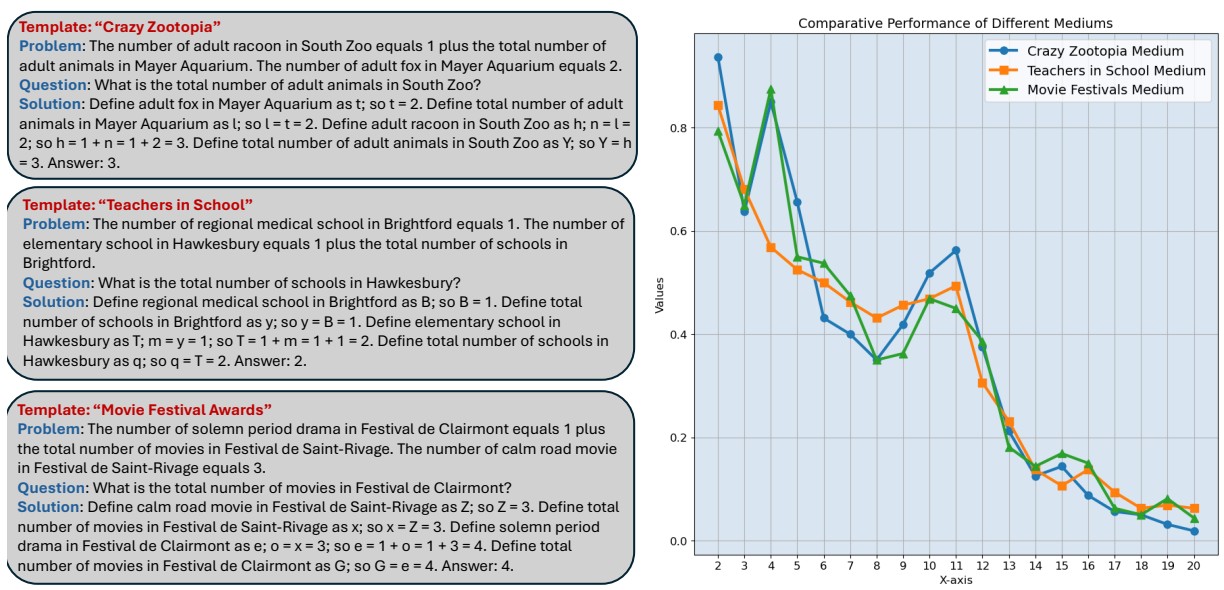

*Figure 12.* Comparison between different task templates

## H. Ablation Study of Graph Structure

To analyze how graph structure impacts model performance, we ran Qwen2.5-7B-Instruct on 40,000 samples from our Symbolic subset with op=9. The Symbolic subset was chosen as it allows us to control for various factors and isolate the impact of graph structure. In this analysis, we focus on the relationship between graph depth and model accuracy. We define the depth of a node as its distance from the root node. The depth of a graph (referred to as "Depth") is determined by the maximum depth of its nodes, while the mean depth represents the average depth across all nodes in the graph. We categorized samples based on their maximum graph depth, segmenting the data into five major categories: Depth=2, Depth=3, Depth=4, Depth=5, and Depth=6. Categories with insufficient data (e.g., fewer than a pre-defined threshold of samples) were excluded to maintain statistical significance. The results, presented below, confirm an inverse relationship: accuracy generally decreases as the maximum graph depth increases, supporting the hypothesis that deeper graphs are more challenging for the model.

| Depth | 2 | 3 | 4 | 5 | 6 |
|---|---|---|---|---|---|
| Accuracy | 0.438 | 0.386 | 0.347 | 0.317 | 0.257 |

Further analysis investigates the relationship between mean depth and accuracy. As shown in Figure 13, there is a clear trend: accuracy tends to decrease as mean depth increases. This inverse relationship suggests that increasing graph complexity, as measured by mean depth, can negatively impact model performance. Notably, when comparing graph groups with similar mean depths, the maximum depth (Depth category) appears to have a less significant impact on accuracy. This finding implies that it is the overall complexity represented by the mean depth, rather than the specific maximum depth of the longest path, that primarily influences performance.

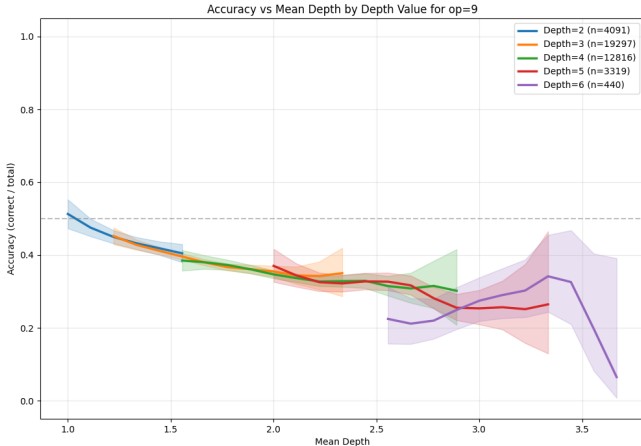

*Figure 13.* Trend of accuracy versus mean depth, categorized by maximum graph depth (Depth). Error bars indicate 95% confidence intervals.

## I. Detailed Model Comparisons

Here we present and highlight experiment results, potentially complementing those in Table 1 and following the discussion of Section 5. We focus on three main aspects of inter-model comparisons.

First, reasoning-enhanced models demonstrate a significant performance advantage over their non-reasoning counterparts, even with identical parameter counts and model architectures. The following table illustrates this comparison across different operational complexities ('op'):

| op | DeepSeek R1 | DeepSeek V3 | Qwen QWQ 32B Preview | Qwen 2.5 32B Instruct (non-reasoning) |
|---|---|---|---|---|
| 10 | 0.9453 | 0.7850 | 0.6400 | 0.6100 |
| 20 | 0.9375 | 0.6000 | 0.5950 | 0.3600 |
| 30 | 0.9453 | 0.3550 | 0.2550 | 0.1200 |
| 40 | 0.9231 | 0.1625 | 0.1250 | 0.0437 |
| **AUC** | 8573.800 | 2407.855 | 1846.187 | 1399.805 |

Second, within the same model family, larger models generally exhibit higher performance than smaller models. This trend is evident in the comparisons below for Qwen and Llama families:

| op | Qwen 2.5 72B Instruct | Qwen 2.5 7B Instruct | Llama 3.1 70B Instruct | Llama 3.1 8B Instruct |
|---|---|---|---|---|
| 5 | 0.9500 | 0.5850 | 0.7300 | 0.5200 |
| 10 | 0.8150 | 0.2850 | 0.6400 | 0.3950 |
| 15 | 0.7550 | 0.1800 | 0.4900 | 0.1900 |
| 20 | 0.6400 | 0.0500 | 0.3250 | 0.0700 |
| 25 | 0.4850 | 0.0250 | 0.2625 | 0.0600 |
| 30 | 0.3650 | 0.0000 | 0.0375 | 0.0200 |
| **AUC** | 2016.375 | 618.500 | 1205.250 | 606.500 |

Third, the results also indicate a performance gap for models with hybrid architectures when compared to those utilizing a standard Transformer architecture. The following table details these findings, comparing Area Under Curve (AUC) scores:

**GSM-∞: How Do your LLMs Behave over Infinitely Increasing Reasoning Complexity and Context Length?**

| Models | Architecture | Date of release | Parameter Size | AUC |
|---|---|---|---|---|
| Qwen-2.5-32B-Instruct | Transformer | 09/17/2024 | 32B | 1405.055 |
| MiniMax-Text-01 | Hybrid (Linear Attention) | 01/14/2025 | 456B MoE (45.9B per token) | 1178.510 |
| Jamba-1.5-Large | Hybrid (SSM) | 08/22/2024 | 398B MoE (98B per token) | 466.400 |

## J. Cost Estimation

We evaluate at least 100 examples for each operational complexity ('op') setting in GSM-Infinite across our three distinct subsets: Symbolic, Medium, and Hard. To illustrate our evaluation scope, consider Llama-3.1-70B-Instruct. We aim to capture the full performance spectrum, from 'op = 2' (lowest reasoning complexity) until the model's accuracy drops below 0.05. This typically involves evaluating up to 40 different 'op' settings for each subset; specifically, the Symbolic subset often requires up to 55 'op' settings, the Medium subset up to 45, and the Hard subset up to 30.

Our sampling strategy for 'op' values varies:

- **Zero-context evaluations:** For most models, we evaluate every 'op' from 2 to 30. Beyond 'op = 30', where accuracy for models around 70B parameters typically decays significantly, we evaluate one 'op' setting per five consecutive 'op' values, adjusting this based on observed model performance.

- **Longer context evaluations:** To manage inference time and cost, we evaluate every second 'op' setting within the first 30 'op' values.

Across the subsets, this translates to approximately 35 evaluation runs (distinct 'op' settings) for the Symbolic subset, 33 runs for Medium, and 30 runs for Hard. For all context length settings, we allow up to 4000 tokens for generation, though for many API-based reasoning models, this output limit cannot be explicitly specified. For zero-context evaluations, the input length is typically upper-bounded at 1000 tokens per example.

Under a worst-case estimate for zero-context evaluations, the total number of examples would be $(35+33+30)\times100 = 9800$ examples. Based on this, we estimate approximately 9.8 million input tokens and 39.2 million output tokens for the zero-context setting. Using representative API costs for a 70B model (e.g., \$0.12 per million input tokens and \$0.30 per million output tokens, noting that local deployment can be virtually free of direct API costs), we summarize the estimated costs for evaluating Llama-3.1-70B-Instruct across different context lengths in Table 15.

*Table 15.* Estimated Evaluation Cost for Llama-3.1-70B-Instruct.

| Context Length | Input Tokens (M) | Output Tokens (M) | Input Cost ($) | Output Cost ($) | Total Cost ($) |
|---|---|---|---|---|---|
| Zero Context | 9.8 | 39.2 | 1.18 | 11.76 | 12.94 |
| 8K | 21.6 | 10.8 | 2.59 | 3.24 | 5.83 |
| 16K | 38.4 | 9.6 | 4.61 | 2.88 | 7.49 |
| 32K | 80.0 | 10.0 | 9.60 | 3.00 | 12.60 |
| **Total** | | | | | **38.86** |

Our experiments consistently show that larger reasoning models are generally more costly to evaluate, while smaller or less complex models are less expensive.

Furthermore, we investigated the effectiveness of using strides in 'op' selection and reducing the number of samples per 'op' to potentially lower evaluation costs without significantly impacting the Area Under Curve (AUC) metric. For this study, we used the zero-context Hard subset and Qwen-2.5-32B-Instruct, which required 42 runs (from 'op = 2' to 'op = 42') for its performance to saturate. We evaluated scenarios with different stride values, as shown in Table 16. The results indicate that strides up to 4 (evaluating every 5th 'op' setting) introduce only a minor effect on the overall AUC.

We typically use 200 samples per 'op' for our standard measurements. To explore further cost reduction, we simulated scenarios with 100 and 50 samples per 'op' using probability bootstrapping. As shown in Table 17, using fewer samples can sometimes yield similar AUC values, offering another avenue for reducing evaluation costs.

*Table 16.* Effect of Stride in 'op' Selection on AUC (Qwen-2.5-32B-Instruct, Zero-Context Hard Subset).

| Stride Configuration | AUC | Difference (%) |
|---|---|---|
| Stride = 0 (evaluating all ops) | 1405.1 | |
| Stride = 0 (ops 2-30), Stride = 5 (ops >30) | 1401.6 | -0.25 |
| Stride = 1 (evaluating every 2nd op) | 1413.1 | 0.57 |
| Stride = 2 (evaluating every 3rd op) | 1382.3 | -1.62 |
| Stride = 3 (evaluating every 4th op) | 1404.99 | -0.01 |
| Stride = 4 (evaluating every 5th op) | 1369.5 | -2.53 |

*Table 17.* Effect of Sample Size per 'op' on AUC (Simulated using Bootstrapping).

| Samples Taken per 'op' | AUC | Difference (%) |
|---|---|---|
| 200 | 1405.1 | |
| 100 | 1419.0 | 0.99 |
| 50 | 1387.0 | -1.29 |

## K. Impact of Problem Structure: Forward vs. Reverse Reasoning

The distinction between forward and reverse reasoning problems in our dataset does not primarily concern the explicit arithmetic operations used (e.g., plus, minus, multiply, divide), as these appear in all subsets (referencing Section 4.3). Instead, the core difference lies in the implicit operations and relationships embedded within the problem structure. As illustrated in Figure 6, implicit addition can represent class-instance relationships, while implicit multiplication can represent hierarchical instance dependencies.

Forward problems exclusively contain these two types of implicit operations (constructive dependencies). Consequently, their computation graphs typically show a flow where more concrete, detailed instances are computed first, and these are then aggregated to determine more general or summary variables.

In contrast, reverse problems involve implicit "minusing" or "divisions." In these scenarios, values for more abstract variables are often provided, and the language model must deduce unknown instance variables. Consider the following simplified examples:

**Forward Problem Example:** The number of pigs in Jefferson Park is 2. The number of sheeps in Jefferson Park is 4. Assume animals not mentioned for this location are not present. What is the total number of animals in Jefferson Park? *Solution Path: Number of pigs (instance) + Number of sheeps (instance) → Total animals (abstract).*

**Reverse Problem Example:** The number of pigs in Jefferson Park is 2. Sheep are present in Jefferson Park in a non-zero quantity. Assume animals not mentioned for this location are not present. The total number of animals in Jefferson Park is 6. How many sheep are in Jefferson Park? *Solution Path: Total animals (abstract) - Number of pigs (instance) → Number of sheeps (instance).*

As detailed in Appendix F, most evaluated LLMs, with some exceptions, achieve better scores on forward problems than on reverse ones, given the same level of reasoning difficulty (i.e., the same number of operations, 'ops'). We hypothesize that this disparity stems from the nature of human-generated text; humans naturally tend to write and explain concepts in the constructive logical ordering characteristic of forward problems. This could lead to a relative scarcity of training data exemplifying reverse logic compared to forward logic.

To investigate whether targeted training can mitigate this gap, we conducted experiments with Qwen-2.5-500M-Instruct. We prepared a dataset of 1.3 million diverse problems, ranging from 'op=2' to 'op=30', generated using our proposed generator (with templates differing from the test set). This training set was balanced, with an equal number of forward and reverse problems (650,000 each).

Our findings indicate that the off-the-shelf 500M model initially demonstrates better forward reasoning ability within its effective performance range (accuracy ≥ 0.05). Beyond this range, its responses tend towards guessing. Interestingly, for reverse problems, due to a potentially smaller search space for the queried variable, the model might exhibit a slightly higher

probability of correctly guessing in this guessing regime, leading to a deceptive slight increase in reverse problem accuracy at higher 'ops'.

After fine-tuning on the 1.3M example dataset, the 500M model showed significant performance improvements on GSM-Infinite. Notably, its performance on reverse problems became slightly higher than on forward problems, suggesting that providing sufficient relevant training data can substantially enhance the model's reasoning capabilities and largely erase the observed forward-reverse performance gap. The results are summarized in Table 18.

*Table 18.* Performance Comparison of Qwen-2.5-500M-Instruct Before and After Fine-tuning on Forward and Reverse Problems.

| Evaluations (op) | Qwen2.5 500M Instruct | | | Qwen2.5 500M Instruct Finetuned with 1.3M Examples | | |
|---|---|---|---|---|---|---|
| | **Forward** | **Reverse** | **Score** | **Forward** | **Reverse** | **Score** |
| 2 | 0.2128 | 0.1385 | 0.1612 | 0.3032 | 0.5846 | 0.5009 |
| 4 | 0.1093 | 0.0743 | 0.0855 | 0.6721 | 0.4505 | 0.5915 |
| 8 | 0.0302 | 0.0491 | 0.0405 | 0.6604 | 0.7791 | 0.7547 |
| 10 | 0.0084 | 0.0390 | 0.0311 | 0.6681 | 0.6780 | 0.7076 |
| 15 | 0.0000 | 0.0690 | 0.0346 | 0.5923 | 0.6767 | 0.6451 |
| 20 | 0.0000 | 0.0698 | 0.0332 | 0.5207 | 0.5488 | 0.5206 |
| 25 | 0.0000 | 0.0533 | 0.0317 | 0.3413 | 0.3413 | 0.3506 |

# L. Long-Context Degradation of Models

In this section, we provide the accuracy decay curves of all LLMs tested across zero-context, 8K, 16K and 32K for further analysis. We selected the first 30 reasoning steps to truncate the data for comparison purposes.

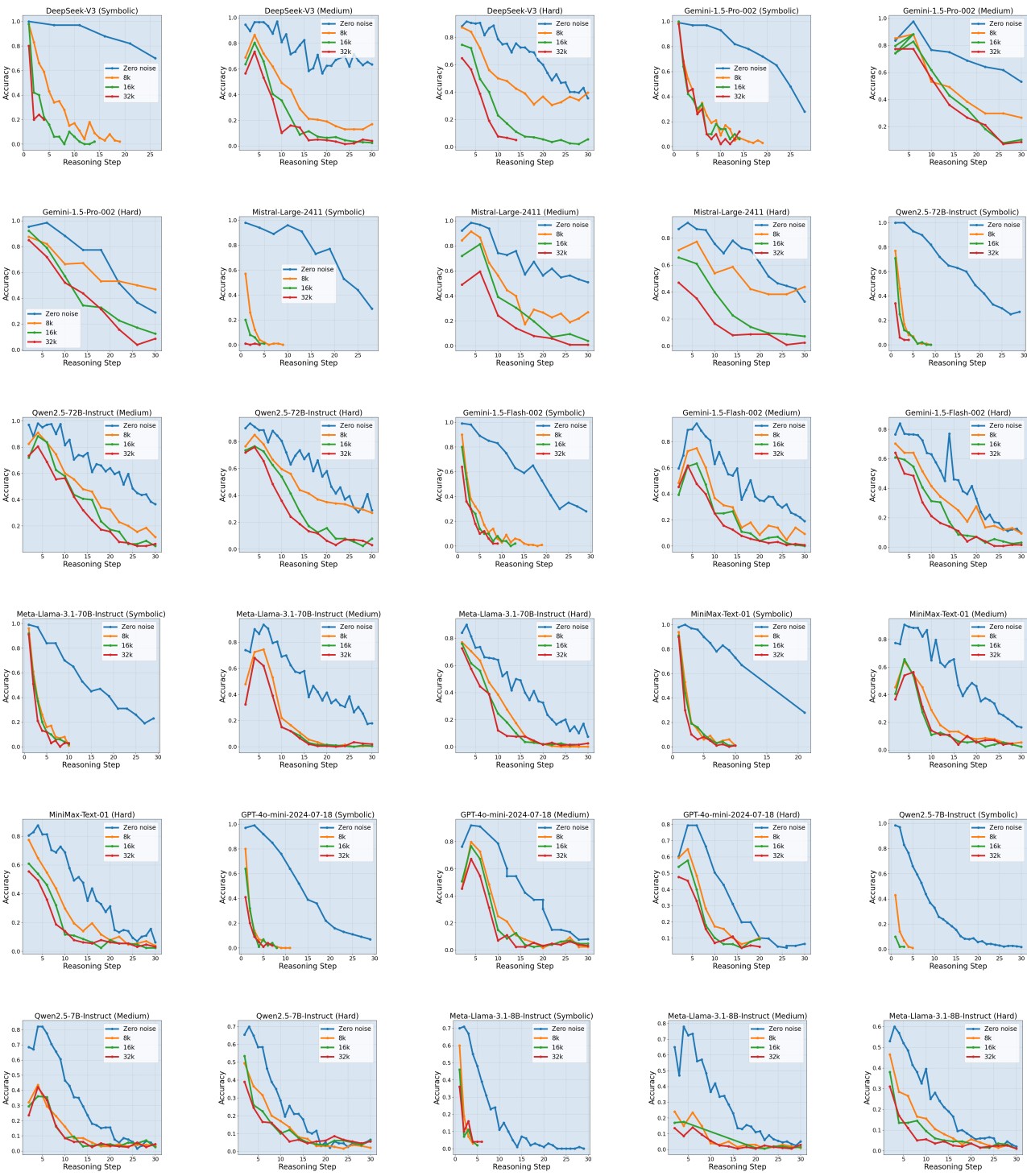

*Figure 14.* Accuracy decay with context length for different models

## M. Common Error Analysis

In this section, we collect and analyze representative examples of errors made by Llama-3.1-8B-Instruct and Qwen-2.5-7B-Instruct. We present ten examples, labeled from 1 to 10. The following table summarizes common error categories.

**Summary of Common Error Types**

| Error Type | Index of Examples |
|---|---|
| Confuses two similar but distinctive concepts in the question | (1), (8), (10) |
| Hallucinates a concept not mentioned in the question | (2), (4), (9) |
| Gets distracted by unnecessary variables in the question | (3) |
| Misinterprets the relationship mentioned in the question | (5), (6), (7) |

*Table 19.* Common error types observed and their corresponding example indices.

**Example 1: Llama-3.1-8B-Instruct**

**Error Description:** Llama-3.1-8B-Instruct confuses two similar concepts within deduction: "Total number of adult animals in Mayer Aquarium" and "Total number of newborn animal children in Mayer Aquarium", which are very distinct concepts.

QUESTION

```
Problem: The average number of newborn children per adult crow in South Zoo equals 1 plus
    the sum of the total number of newborn animal children in Mayer Aquarium and the
    number of adult eagle in Mayer Aquarium. The number of adult eagle in Mayer Aquarium
    equals the number of adult owl in Mayer Aquarium. The number of adult parrot in South
    Zoo equals 4. The average number of newborn children per adult owl in Jefferson
    Circus equals the sum of the total number of newborn animal children in Mayer
    Aquarium, the total number of adult animals in Mayer Aquarium, and the average number
    of newborn children per adult owl in Mayer Aquarium. The number of adult owl in Mayer
    Aquarium equals 3. The number of adult eagle in Jefferson Circus equals 4 times the
    sum of the number of adult parrot in South Zoo, the number of adult owl in Bundle
    Ranch, and the average number of newborn children per adult owl in Bundle Ranch. The
    average number of newborn children per adult eagle in Jefferson Circus equals 4. The
    number of adult crow in South Zoo equals the average number of newborn children per
    adult eagle in Mayer Aquarium. The average number of newborn children per adult
    parrot in South Zoo equals 3. The number of adult owl in Bundle Ranch equals 2 plus
    the number of adult parrot in South Zoo. The average number of newborn children per
    adult crow in Jefferson Circus equals the average number of newborn children per
    adult owl in Jefferson Circus. The number of adult owl in Jefferson Circus equals 4.
    The average number of newborn children per adult owl in Bundle Ranch equals the
    difference between the number of adult owl in Bundle Ranch and the number of adult
    parrot in South Zoo. The average number of newborn children per adult owl in Mayer
    Aquarium equals 3. The average number of newborn children per adult eagle in Mayer
    Aquarium equals the total number of adult animals in Mayer Aquarium. The number of
    adult crow in Jefferson Circus equals the average number of newborn children per
    adult owl in Bundle Ranch.
Question: What is the average number of newborn children per adult eagle in Mayer
    Aquarium?
```

GROUND TRUTH SOLUTION

```
Define adult owl in Mayer Aquarium as B; so B = 3.
Define adult eagle in Mayer Aquarium as y; so y = B = 3.
Define total number of adult animals in Mayer Aquarium as N; so N = y + B = 3 + 3 = 6.
Define average number of newborn children per adult eagle in Mayer Aquarium as k; so k =
    N = 6.
Answer: 6.
```

LLAMA-3.1-8B-INSTRUCT'S OUTPUT

```
to solve this problem, we need to break it down into smaller steps and use the given
    information to calculate the average number of newborn children per adult eagle in
    mayer aquarium.

step 1: calculate the total number of newborn animal children in mayer aquarium.

from the problem statement, we know that the average number of newborn children per adult
    owl in mayer aquarium equals 3. since the number of adult owl in mayer aquarium
    equals 3, the total number of newborn animal children in mayer aquarium equals 3 * 3
    = 9.

step 2: calculate the total number of adult animals in mayer aquarium.

from the problem statement, we know that the average number of newborn children per adult
    eagle in mayer aquarium equals the total number of adult animals in mayer aquarium.
    let's call the total number of adult animals in mayer aquarium as x. then, the
    average number of newborn children per adult eagle in mayer aquarium equals x.

step 3: calculate the average number of newborn children per adult eagle in mayer
    aquarium.

from the problem statement, we know that the average number of newborn children per adult
    eagle in mayer aquarium equals the total number of adult animals in mayer aquarium.
    therefore, we can write an equation:

x = x

this equation is true for any value of x. however, we need to find the value of x.
step 4: use the information from the problem statement to find the value of x.
from the problem statement, we know that the average number of newborn children per adult
    eagle in mayer aquarium equals the total number of newborn animal children in mayer
    aquarium. we already calculated the total number of newborn animal children in mayer
    aquarium as 9. therefore, we can write an equation:
x = 9
now we have a value for x.
step 5: calculate the average number of newborn children per adult eagle in mayer
    aquarium.
since x = 9, the average number of newborn children per adult eagle in mayer aquarium
    equals 9.
answer: 9
```

## Example 2: Llama-3.1-8B-Instruct

**Error Description:** Llama-3.1-8B-Instruct makes up a concept that isn't appeared in the question statement. "Adult Parrot in Hamilton Farm" is introduced in the response, which is not mentioned in the question. The newly introduced concept cannot be computed based on the question, since it is not mentioned there, and Llama-3.1-8B-Instruct is stuck at computing it.

QUESTION

```
Problem: The average number of newborn children per adult blue jay in Bundle Ranch equals
    2. The number of adult parrot in Bundle Ranch equals 2. The number of adult blue jay
    in Bundle Ranch equals 2 times the average number of newborn children per adult blue
    jay in Bundle Ranch. The number of adult eagle in Bundle Ranch equals 2 times the
    average number of newborn children per adult blue jay in Bundle Ranch. The number of
    adult parrot in South Zoo equals 4 times the sum of the average number of newborn
    children per adult eagle in Hamilton Farm, the number of adult eagle in Hamilton
    Farm, and the average number of newborn children per adult eagle in Hamilton Farm.
    The average number of newborn children per adult eagle in Hamilton Farm equals the
    number of adult eagle in Bundle Ranch. The number of adult eagle in Hamilton Farm
    equals 3. The average number of newborn children per adult parrot in Bundle Ranch
    equals the total number of adult animals in Hamilton Farm. The number of adult eagle
```

in South Zoo equals 1. The average number of newborn children per adult parrot in
South Zoo equals the average number of newborn children per adult parrot in Bundle
Ranch. The average number of newborn children per adult eagle in Bundle Ranch equals
3 plus the average number of newborn children per adult parrot in Bundle Ranch. The
average number of newborn children per adult eagle in South Zoo equals the sum of the
number of adult blue jay in Bundle Ranch, the average number of newborn children per
adult blue jay in Bundle Ranch, the average number of newborn children per adult
parrot in Bundle Ranch, and the number of adult parrot in Bundle Ranch. Question:
What is the average number of newborn children per adult eagle in Bundle Ranch?

## GROUND TRUTH SOLUTION

Solution: Define adult eagle in Hamilton Farm as H; so H = 3. Define total number of
adult animals in Hamilton Farm as w; so w = H = 3. Define average number of newborn
children per adult parrot in Bundle Ranch as A; so A = w = 3. Define average number
of newborn children per adult eagle in Bundle Ranch as a; U = A = 3; so a = 3 + U = 3
+ 3 = 6. Answer: 6.

## LLAMA-3.1-8B-INSTRUCT'S OUTPUT

to find the average number of newborn children per adult eagle in bundle ranch, we need
to follow the given information step by step.
step 1: the number of adult blue jay in bundle ranch equals 2 times the average number of
newborn children per adult blue jay in bundle ranch.
let's denote the average number of newborn children per adult blue jay in bundle ranch as
x. then, the number of adult blue jay in bundle ranch is 2x.
step 2: the number of adult eagle in bundle ranch equals 2 times the average number of
newborn children per adult blue jay in bundle ranch.
since the average number of newborn children per adult blue jay in bundle ranch is x, the
number of adult eagle in bundle ranch is 2x.
step 3: the average number of newborn children per adult eagle in bundle ranch equals 3
plus the average number of newborn children per adult parrot in bundle ranch.
let's denote the average number of newborn children per adult parrot in bundle ranch as
y. then, the average number of newborn children per adult eagle in bundle ranch is 3
+ y.
step 4: the average number of newborn children per adult parrot in bundle ranch equals
the total number of adult animals in hamilton farm.
we need to find the total number of adult animals in hamilton farm. to do this, we need
to find the number of adult eagle in hamilton farm and the number of adult parrot in
hamilton farm.
step 5: the number of adult eagle in hamilton farm equals 3.
the average number of newborn children per adult eagle in hamilton farm equals the number
of adult eagle in bundle ranch.
since the number of adult eagle in bundle ranch is 2x, the average number of newborn
children per adult eagle in hamilton farm is also 2x.
step 6: the average number of newborn children per adult parrot in bundle ranch equals
the total number of adult animals in hamilton farm.
the total number of adult animals in hamilton farm is the sum of the number of adult
eagle in hamilton farm and the number of adult parrot in hamilton farm. since the
number of adult eagle in hamilton farm is 3, and the average number of newborn
children per adult eagle in hamilt
on farm is 2x, the number of adult parrot in hamilton farm is (2x − 3) / y.
step 7: the number of adult parrot in south zoo equals 4 times the sum of the average
number of newborn children per adult eagle in hamilton farm, the number of adult
eagle in hamilton farm, and the average number of newborn children per adult eagle in
hamilton farm.
the sum of the average number of newborn children per adult eagle in hamilton farm, the
number of adult eagle in hamilton farm, and the average number of newborn children
per adult eagle in hamilton farm is 2x + 3 + 2x = 4x + 3. therefore, the number of
adult parrot in south zoo
is 4(4x + 3).
step 8: the average number of newborn children per adult parrot in south zoo equals the
average number of newborn children per adult parrot in bundle ranch.

```
the average number of newborn children per adult parrot in bundle ranch is y. therefore,
    the average number of newborn children per adult parrot in south zoo is also y.
step 9: the average number of newborn children per adult eagle in south zoo equals the
    sum of the number of adult blue jay in bundle ranch, the average number of newborn
    children per adult blue jay in bundle ranch, the average number of newborn children
    per adult parrot in bundle
ranch, and the number of adult parrot in bundle ranch.
the sum of the number of adult blue jay in bundle ranch, the average number of newborn
    children per adult blue jay in bundle ranch, the average number of newborn children
    per adult parrot in bundle ranch, and the number of adult parrot in bundle ranch is
    2x + x + y + 2x = 4x + x + y = 5x + y.
```

### Example 3: Llama-3.1-8B-Instruct

**Error Description:** Llama-3.1-8B-Instruct already compute the correct answer. However, when it re-evaluating itself, it decides that it needs to use unnecessary variables into the calculation by introducing un-mentioned relationship from the question, and Llama-3.1-8B-Instruct arrives at the wrong result.

QUESTION

```
Problem: The average number of newborn children per adult blue jay in Oakridge Riverside
    equals 2. The average number of newborn children per adult parrot in Cedar Valley
    equals 1 plus the total number of adult animals in Oakridge Riverside. The number of
    adult parrot in Cedar Valley equals the number of adult blue jay in Cedar Valley. The
    number of adult blue jay in Oakridge Riverside equals the average number of newborn
    children per adult blue jay in Oakridge Riverside. The average number of newborn
    children per adult blue jay in Cedar Valley equals 4. The number of adult blue jay in
    Cedar Valley equals 3 times the sum of the average number of newborn children per
    adult blue jay in Oakridge Riverside, the total number of adult animals in Oakridge
    Riverside, and the number of adult blue jay in Oakridge Riverside. Question: What is
    the average number of newborn children per adult parrot in Cedar Valley?
```

GROUND TRUTH SOLUTION

```
Define average number of newborn children per adult blue jay in Oakridge Riverside as I;
    so I = 2. Define adult blue jay in Oakridge Riverside as V; so V = I = 2. Define
    total number of adult animals in Oakridge Riverside as K; so K = V = 2. Define
    average number of newborn children per adult parrot in Cedar Valley as O; h = K = 2;
    so O = 1 + h = 1 + 2 = 3. Answer: 3.
```

LLAMA-3.1-8B-INSTRUCT'S OUTPUT

```
to solve this problem, we need to break it down into smaller steps and use the given
    information to calculate the required quantities.
**step 1: calculate the number of adult blue jays in oakridge riverside**
from the problem statement, we know that the number of adult blue jays in oakridge
    riverside equals the average number of newborn children per adult blue jay in
    oakridge riverside. since the average number of newborn children per adult blue jay
    in oakridge riverside equals 2, we can conclude that the number of adult blue jays in
    oakridge riverside is also 2.
**step 2: calculate the total number of adult animals in oakridge riverside**
we already know that the number of adult blue jays in oakridge riverside is 2. since
    there is no other type of animal mentioned for oakridge riverside, we can assume that
    the total number of adult animals in oakridge riverside is also 2.
**step 3: calculate the average number of newborn children per adult parrot in cedar
    valley**
from the problem statement, we know that the average number of newborn children per adult
    parrot in cedar valley equals 1 plus the total number of adult animals in oakridge
    riverside. we already calculated that the total number of adult animals in oakridge
    riverside is 2. therefore, the average number of newborn children per adult parrot in
    cedar valley equals 1 + 2 = 3.
**step 4: calculate the number of adult parrots in cedar valley**
```

```
from the problem statement, we know that the number of adult parrots in cedar valley
    equals the number of adult blue jays in cedar valley. we don't know the number of
    adult blue jays in cedar valley yet, so we'll come back to this later.
**step 5: calculate the number of adult blue jays in cedar valley**
from the problem statement, we know that the number of adult blue jays in cedar valley
    equals 3 times the sum of the average number of newborn children per adult blue jay
    in oakridge riverside, the total number of adult animals in oakridge riverside, and
    the number of adult blue jays in oakridge riverside. we already calculated that the
    average number of newborn children per adult blue jay in oakridge riverside is 2, the
    total number of adult animals in oakridge riverside is 2, and the number of adult
    blue jays in oakridge riverside is 2. therefore, the number of adult blue jays in
    cedar valley equals 3 * (2 + 2 + 2) = 3 * 6 = 18.
**step 6: calculate the number of adult parrots in cedar valley**
from the problem statement, we know that the number of adult parrots in cedar valley
    equals the number of adult blue jays in cedar valley. we already calculated that the
    number of adult blue jays in cedar valley is 18. therefore, the number of adult
    parrots in cedar valley is also 18.
**step 7: calculate the average number of newborn children per adult parrot in cedar
    valley**
from the problem statement, we know that the average number of newborn children per adult
    parrot in cedar valley equals 1 plus the total number of adult animals in oakridge
    riverside. we already calculated that the total number of adult animals in oakridge
    riverside is 2. therefore, the average number of newborn children per adult parrot in
    cedar valley equals 1 + 2 = 3.
however, we need to re-evaluate this step. the average number of newborn children per
    adult parrot in cedar valley is actually given by the problem statement as 1 plus the
    total number of adult animals in oakridge riverside. but we also know that the number
    of adult parrots in cedar valley equals the number of adult blue jays in cedar
    valley, which is 18. therefore, the average number of newborn children per adult
    parrot in cedar valley equals 1 + (total number of adult animals in oakridge
    riverside) + (number of adult blue jays in cedar valley / number of adult parrots in
    cedar valley) * (average number of newborn children per adult blue jay in cedar
    valley - average number of newborn children per adult parrot in cedar valley). we
    already calculated that the total number of adult animals in oakridge riverside is 2,
    the number of adult blue jays in cedar valley is 18, and the average number of
    newborn children per adult blue jay in cedar valley is 4. therefore, the average
    number of newborn children per adult parrot in cedar valley equals 1 + 2 + (18/18) *
    (4 - 3) = 1 + 2 + 1 = 4.
answer: 4
```

### Example 4: Llama-3.1-8B-Instruct

**Error Description:** Llama-3.1-8B-Instruct make up a concept "adult bear in Pine Ridge" that isn't mentioned in the question. The response cannot compute the correct answer from a concept that cannot be computed which is introduced earlier in the response.

QUESTION

```
Problem: The average number of newborn children per adult bear in Maple Creek equals 4.
    The number of adult bear in Beverly Forest equals 2. The average number of newborn
    children per adult bear in Beverly Forest equals 3. The average number of newborn
    children per adult wolf in Pine Ridge equals the number of adult bear in Cedar
    Valley. The number of adult wolf in Pine Ridge equals 4. The number of adult racoon
    in Pine Ridge equals 3 times the number of adult wolf in Pine Ridge. The number of
    adult wolf in Maple Creek equals 4. The average number of newborn children per adult
    racoon in Beverly Forest equals the sum of the number of adult racoon in Beverly
    Forest, the average number of newborn children per adult bear in Beverly Forest, and
    the number of adult wolf in Maple Creek. The average number of newborn children per
    adult wolf in Maple Creek equals 4 times the sum of the average number of newborn
    children per adult bear in Beverly Forest and the average number of newborn children
    per adult bear in Cedar Valley. The number of adult wolf in Beverly Forest equals the
    sum of the total number of adult animals in Pine Ridge and the number of adult wolf
    in Pine Ridge. The number of adult racoon in Beverly Forest equals the average number
```

of newborn children per adult bear in Maple Creek. The average number of newborn
children per adult racoon in Pine Ridge equals the total number of adult animals in
Maple Creek. The number of adult bear in Maple Creek equals the number of adult bear
in Cedar Valley. The number of adult bear in Cedar Valley equals 2. The average
number of newborn children per adult bear in Cedar Valley equals the sum of the
average number of newborn children per adult bear in Beverly Forest and the average
number of newborn children per adult bear in Maple Creek. The average number of
newborn children per adult wolf in Beverly Forest equals the average number of
newborn children per adult bear in Maple Creek. Question: What is the total number of
adult animals in Pine Ridge?

GROUND TRUTH SOLUTION

Define adult wolf in Pine Ridge as v; so v = 4. Define adult racoon in Pine Ridge as R; r
    = v = 4; so R = 3 * r = 3 * 4 = 12. Define total number of adult animals in Pine
    Ridge as b; so b = v + R = 4 + 12 = 16. Answer: 16.

LLAMA-3.1-8B-INSTRUCT'S OUTPUT

to solve this problem, we need to break it down into smaller steps and use the given
    information to calculate the total number of adult animals in pine ridge.
step 1: calculate the number of adult bear in cedar valley.
the number of adult bear in cedar valley equals 2.
step 2: calculate the average number of newborn children per adult bear in cedar valley.
the average number of newborn children per adult bear in cedar valley equals the sum of
    the average number of newborn children per adult bear in beverly forest and the
    average number of newborn children per adult bear in maple creek.
let's denote the average number of newborn children per adult bear in beverly forest as x
    and the average number of newborn children per adult bear in maple creek as 4.
the average number of newborn children per adult bear in cedar valley = x + 4
step 3: calculate the average number of newborn children per adult wolf in pine ridge.
the average number of newborn children per adult wolf in pine ridge equals the number of
    adult bear in cedar valley.
the average number of newborn children per adult wolf in pine ridge = 2
step 4: calculate the number of adult wolf in pine ridge.
the number of adult wolf in pine ridge equals 4.
step 5: calculate the number of adult racoon in pine ridge.
the number of adult racoon in pine ridge equals 3 times the number of adult wolf in pine
    ridge.
the number of adult racoon in pine ridge = 3 * 4 = 12
step 6: calculate the total number of adult animals in pine ridge.
the total number of adult animals in pine ridge equals the sum of the number of adult
    wolf, the number of adult racoon, and the number of adult bear in pine ridge.
however, the number of adult bear in pine ridge is not given, so we need to calculate it.
step 7: calculate the average number of newborn children per adult racoon in pine ridge.
the average number of newborn children per adult racoon in pine ridge equals the total
    number of adult animals in maple creek.
however, the total number of adult animals in maple creek is not given, so we need to
    calculate it.
step 8: calculate the total number of adult animals in maple creek.
the total number of adult animals in maple creek equals the sum of the number of adult
    bear, the number of adult wolf, and the number of adult racoon in maple creek.
however, the number of adult racoon in maple creek is not given, so we need to calculate
    it.
step 9: calculate the number of adult racoon in maple creek.
the number of adult racoon in maple creek equals the average number of newborn children
    per adult bear in maple creek.
however, the average number of newborn children per adult bear in maple creek equals 4.
step 10: calculate the number of adult racoon in maple creek.
the number of adult racoon in maple creek = 4
step 11: calculate the total number of adult animals in maple creek.
the total number of adult animals in maple creek equals the sum of the number of adult
    bear, the number of adult wolf, and the number of adult racoon in maple creek.

the number of adult bear in maple creek equals the number of adult bear in cedar valley,
    which equals 2.
the number of adult wolf in maple creek equals 4.
the total number of adult animals in maple creek = 2 + 4 + 4 = 10
step 12: calculate the average number of newborn children per adult racoon in pine ridge.
the average number of newborn children per adult racoon in pine ridge equals the total
    number of adult animals in maple creek.
the average number of newborn children per adult racoon in pine ridge = 10
step 13: calculate the number of adult racoon in pine ridge.
the number of adult racoon in pine ridge equals 3 times the number of adult wolf in pine
    ridge.
however, we already calculated the number of adult racoon in pine ridge in step 5, which
    equals 12.
step 14: calculate the total number of adult animals in pine ridge.
the total number of adult animals in pine ridge equals the sum of the number of adult
    wolf, the number of adult racoon, and the number of adult bear in pine ridge.
the number of adult wolf in pine ridge equals 4.
the number of adult racoon in pine ridge equals 12.
however, the number of adult bear in pine ridge is not given, so we need to calculate it.
step 15: calculate the average number of newborn children per adult wolf in beverly
    forest.
the average number of newborn children per adult wolf in beverly forest equals the
    average number of newborn children per adult bear in mapl
e creek.
the average number of newborn children per adult wolf in beverly forest = 4
step 16: calculate the number of adult wolf in beverly forest.
the number of adult wolf in beverly forest equals the sum of the total number of adult
    animals in pine ridge and the number of adult wolf in
 pine ridge.
however, we already calculated the total number of adult animals in pine ridge in step
    14, which equals 20.
the number of adult wolf in beverly forest = 20 + 4 = 24
step 17: calculate the number of adult bear in beverly forest.
the average number of newborn children per adult bear in beverly forest equals 3.
the total number of newborn children per adult bear in beverly forest equals the average
    number of newborn children per adult bear in beverly forest times the number of adult
    bear in beverly forest.
however, the number of adult bear in beverly forest is not given, so we need to calculate
    it.
step 18: calculate the number of adult bear in beverly forest.
the number of adult bear in beverly forest equals 2.
step 19: calculate the total number of newborn children per adult bear in beverly forest.
the total number of newborn children per adult bear in beverly forest equals the average
    number of newborn children per adult bear in beverly forest times the number of adult
    bear in beverly forest.
the total number of newborn children per adult bear in beverly forest = 3 * 2 = 6
step 20: calculate the average number of newborn children per adult bear in cedar valley.
the average number of newborn children per adult bear in cedar valley equals the sum of
    the average number of newborn children per adult bear in beverly forest and the
    average number of newborn children per adult bear in maple creek.
the average number of newborn children per adult bear in cedar valley = 3 + 4 = 7
step 21: calculate the number of adult bear in maple creek.
the number of adult bear in maple creek equals the number of adult bear in cedar valley,
    which equals 2.
step 22: calculate the total number of newborn children per adult bear in maple creek.
the total number of newborn children per adult bear in maple creek equals the average
    number of newborn children per adult bear in maple creek times the number of adult
    bear in maple creek.
the total number of newborn children per adult bear in maple creek = 4 * 2 = 8
step 23: calculate the number of adult racoon in beverly forest.
the number of adult racoon in beverly forest equals the average number of newborn
    children per adult bear in maple creek.
the number of adult racoon in beverly forest = 4
step 24: calculate the average number of newborn children per adult racoon in beverly
    forest.

```
the average number of newborn children per adult racoon in beverly forest equals the sum
    of the number of adult racoon in beverly forest, the average number of newborn
    children per adult bear in beverly forest, and the number of adult wolf in maple
    creek.
the average number of newborn children per adult racoon in beverly forest = 4 + 3 + 4 = 11
step 25: calculate the total number of newborn children per adult racoon in beverly
    forest.
the total number of newborn children per adult racoon in beverly forest equals the
    average number of newborn children per adult racoon in beverly forest times the
    number of adult racoon in beverly forest.
the total number of newborn children per adult racoon in beverly forest = 11 * 4 = 44
step 26: calculate the total number of newborn children per adult racoon in pine ridge.
the total number of newborn children per adult racoon in pine ridge equals the average
    number of newborn children per adult racoon in pine ridge times the number of adult
    racoon in pine ridge.
however, we already calculated the average number of newborn children per adult racoon in
    pine ridge in step 12, which equals 10.
the total number of newborn children per adult racoon in pine ridge = 10 * 12 = 120
step 27: calculate the total number of newborn children per adult wolf in pine ridge.
the total number of newborn children per adult wolf in pine ridge equals the average
    number of newborn children per adult wolf in pine ridge
 times the number of adult wolf in pine ridge.
the total number of newborn children per adult wolf in pine ridge = 2 * 4 = 8
step 28: calculate the total number of newborn children per adult bear in pine ridge.
the total number of newborn children per adult bear in pine ridge equals the average
    number of newborn children per adult bear in cedar valley times the number of adult
    bear in cedar valley.
the total number of newborn children per adult bear in pine ridge = 7 * 2 = 14
step 29: calculate the total number of newborn children in pine ridge.
the total number of newborn children in pine ridge equals the sum of the total number of
    newborn children per adult wolf, the total number of newborn children per adult
    racoon, and the total number of newborn children per adult bear in pine ridge.
the total number of newborn children in pine ridge = 8 + 120 + 14 = 142
step 30: calculate the total number of adult animals in pine ridge.
the total number of adult animals in pine ridge equals the sum of the number of adult
    wolf, the number of adult racoon, and the number of adult bear in pine ridge.
however, the number of adult bear in pine ridge is not given, so we need to calculate it.
step 31: calculate the number of adult bear in pine ridge.
the number of adult bear in pine ridge equals the total number of newborn children per
    adult bear in pine ridge divided by the average number of newborn children per adult
    bear in cedar valley.
the number of adult bear in pine ridge = 14 / 7 = 2
step 32: calculate the total number of adult animals in pine ridge.
the total number of adult animals in pine ridge equals the sum of the number of adult
    wolf, the number of adult racoon, and the number of adult bear in pine ridge.
the total number of adult animals in pine ridge = 4 + 12 + 2 = 18
answer: 18
```

### Example 5: Llama-3.1-8B-Instruct

**Error Description:** Llama-3.1-8B-Instruct misinterpret the condition given in the question. The question states that "average number of newborn children per adult eagle in Maple Creek" equals 2 + "average number of newborn children per adult eagle in Cedar Valley". However, during the responding, Llama-3.1-8B-Instruct compute "average number of newborn children per adult eagle in Maple Creek" as the sum of "average number of newborn children per adult eagle in Cedar Valley" and "average number of newborn children per adult owl in Cedar Valley" which is different from what the question mentions.

QUESTION

```
Problem: The average number of newborn children per adult owl in Cedar Valley equals 4.
    The average number of newborn children per adult eagle in Maple Creek equals 2 plus
    the average number of newborn children per adult eagle in Cedar Valley. The average
    number of newborn children per adult eagle in Cedar Valley equals 2. The number of
```

adult eagle in Maple Creek equals 2 plus the sum of the average number of newborn children per adult owl in Cedar Valley and the number of adult owl in Cedar Valley. The number of adult owl in Oakridge Riverside equals 2 plus the difference between the average number of newborn children per adult eagle in Cedar Valley and the average number of newborn children per adult owl in Oakridge Riverside. The average number of newborn children per adult owl in Oakridge Riverside equals the sum of the average number of newborn children per adult eagle in Maple Creek and the average number of newborn children per adult eagle in Cedar Valley. The number of adult eagle in Cedar Valley equals 3 times the difference between the average number of newborn children per adult eagle in Cedar Valley and the number of adult owl in Cedar Valley. The number of adult owl in Cedar Valley equals the average number of newborn children per adult owl in Cedar Valley. Question: What is the average number of newborn children per adult owl in Oakridge Riverside?

## GROUND TRUTH SOLUTION

Define average number of newborn children per adult eagle in Cedar Valley as u; so u = 2. Define average number of newborn children per adult eagle in Maple Creek as v; M = u = 2; so v = 2 + M = 2 + 2 = 4. Define average number of newborn children per adult owl in Oakridge Riverside as q; so q = v + u = 4 + 2 = 6. Answer: 6.

## LLAMA-3.1-8B-INSTRUCT'S OUTPUT

to solve this problem, we need to carefully follow the given information and use it to find the average number of newborn children per a
dult owl in oakridge riverside.

here are the steps:

1. let's denote the average number of newborn children per adult owl in cedar valley as a, the average number of newborn children per adult eagle in cedar valley as b, the average number of newborn children per adult eagle in maple creek as c, the number of adult eagle in maple creek as d, the number of adult owl in cedar valley as e, the number of adult owl in oakridge riverside as f, and the average number of newborn children per adult owl in oakridge riverside as g.

2. from the problem statement, we know that a = 4.

3. we also know that b = 2.

4. furthermore, we are given that c = b + a = 2 + 4 = 6.

5. the problem also states that d = 2 + (a + e).

6. we are also given that e = a = 4.

7. substituting the value of e into the equation for d, we get d = 2 + (4 + 4) = 10.
8. the problem also states that f = 2 + (b − g).
9. we are also given that g = c + b = 6 + 2 = 8.
10. substituting the value of g into the equation for f, we get f = 2 + (2 − 8) = −4. however, since the number of adult animals cannot
be negative, we need to re-evaluate the equation for f.
11. let's re-evaluate the equation for g: g = c + b = 6 + 2 = 8.
12. now, let's re-evaluate the equation for f: f = 2 + (b − g) = 2 + (2 − 8) = 2 − 6 = −4. however, this is still not correct.
13. let's re-evaluate the equation for f again: f = 2 + (b − g) = 2 + (2 − 8) = 2 − 6 = −4. however, this is still not correct.
14. let's re-evaluate the equation for f once more: f = 2 + (b − g) = 2 + (2 − 8) = 2 − 6 = −4. however, this is still not correct.
15. let's re-evaluate the equation for f again: f = 2 + (b − g) = 2 + (2 − 8) = 2 − 6 = −4. however, this is still not correct.
16. let's re-evaluate the equation for f once more: f = 2 + (b − g) = 2 + (2 − 8) = 2 − 6 = −4. however, this is still not correct.

17. let's re-evaluate the equation for f again: f = 2 + (b - g) = 2 + (2 - 8) = 2 - 6 = -4. however, this is still not correct.
18. let's re-evaluate the equation for f once more: f = 2 + (b - g) = 2 + (2 - 8) = 2 - 6 = -4. however, this is still not correct.
19. let's re-evaluate the equation for f again: f = 2 + (b - g) = 2 + (2 - 8) = 2 - 6 = -4. however, this is still not correct.
20. let's re-evaluate the equation for f once more: f = 2 + (b - g) = 2 + (2 - 8) = 2 - 6 = -4. however, this is still not correct.
21. let's re-evaluate the equation for f again: f = 2 + (b - g) = 2 + (2 - 8) = 2 - 6 = -4. however, this is still not correct.
22. let's re-evaluate the equation for f once more: f = 2 + (b - g) = 2 + (2 - 8) = 2 - 6 = -4. however, this is still not correct.
23. let's re-evaluate the equation for f again: f = 2 + (b - g) = 2 + (2 - 8) = 2 - 6 = -4. however, this is still not correct.
24. let's re-evaluate the equation for f once more: f = 2 + (b - g) = 2 + (2 - 8) = 2 - 6 = -4. however, this is still not correct.
25. let's re-evaluate the equation for f again: f = 2 + (b - g) = 2 + (2 - 8) = 2 - 6 = -4. however, this is still not correct.
26. let's re-evaluate the equation for f once more: f = 2 + (b - g) = 2 + (2 - 8) = 2 - 6 = -4. however, this is still not correct.
27. let's re-evaluate the equation for f again: f = 2 + (b - g) = 2 + (2 - 8) = 2 - 6 = -4. however, this is still not correct.
28. let's re-evaluate the equation for f once more: f = 2 + (b - g) = 2 + (2 - 8) = 2 - 6 = -4. however, this is still not correct.
29. let's re-evaluate the equation for f again: f = 2 + (b - g) = 2 + (2 - 8) = 2 - 6 = -4. however, this is still not correct.
30. let's re-evaluate the equation for f once more: f = 2 + (b - g) = 2 + (2 - 8) = 2 - 6 = -4. however, this is still not correct.
31. let's re-evaluate the equation for f again: f = 2 + (b - g) = 2 + (2 - 8) = 2 - 6 = -4. however, this is still not correct.
32. let's re-evaluate the equation for f once more: f = 2 + (b - g) = 2 + (2 - 8) = 2 - 6 = -4. however, this is still not correct.
33. let's re-evaluate the equation for f again: f = 2 + (b - g) = 2 + (2 - 8) = 2 - 6 = -4. however, this is still not correct.
34. let's re-evaluate the equation for f once more: f = 2 + (b - g) = 2 + (2 - 8) = 2 - 6 = -4. however, this is still not correct.
35. let's re-evaluate the equation for f again: f = 2 + (b - g) = 2 + (2 - 8) = 2 - 6 = -4. however, this is still not correct.
36. let's re-evaluate the equation for f once more: f = 2 + (b - g) = 2 + (2 - 8) = 2 - 6 = -4. however, this is still not correct.
37. let's re-evaluate the equation for f again: f = 2 + (b - g) = 2 + (2 - 8) = 2 - 6 = -4. however, this is still not correct.
38. let's re-evaluate the equation for f once more: f = 2 + (b - g) = 2 + (2 - 8) = 2 - 6 = -4. however, this is still not correct.
39. let's re-evaluate the equation for f again: f = 2 + (b - g) = 2 + (2 - 8) = 2 - 6 = -4. however, this is still not correct.
40. let's re-evaluate the equation for f once more: f = 2 + (b - g) = 2 + (2 - 8) = 2 - 6 = -4. however, this is still not correct.
41. let's re-evaluate the equation for f again: f = 2 + (b - g) = 2 + (2 - 8) = 2 - 6 = -4. however, this is still not correct.
42. let's re-evaluate the equation for f once more: f = 2 + (b - g) = 2 + (2 - 8) = 2 - 6 = -4. however, this is still not correct.
43. let's re-evaluate the equation for f again: f = 2 + (b - g) = 2 + (2 - 8) = 2 - 6 = -4. however, this is still not correct.
44. let's re-evaluate the equation for f once more: f = 2 + (b - g) = 2 + (2 - 8) = 2 - 6 = -4. however, this is still not correct.
45. let's re-evaluate the equation for f again: f = 2 + (b - g) = 2 + (2 - 8) = 2 - 6 = -4. however, this is still not correct.
46. let's re-evaluate the equation for f once more: f = 2 + (b - g) = 2 + (2 - 8) = 2 - 6 = -4. however, this is still not correct.
47. let's re-evaluate the equation for f again: f = 2 + (b - g) = 2 + (2 - 8) = 2 - 6 = -4. however, this is still not correct.
48. let's re-evaluate the equation for f once more: f = 2 + (b - g) = 2 + (2 - 8) = 2 - 6 = -4. however, this is still not correct.
49. let's re-evaluate the equation for f again: f = 2 + (b - g) = 2 + (2 - 8) = 2 - 6 =

-4. however, this is still not correct.
50. let's re-evaluate the equation for f once more: f = 2 + (b - g) = 2 + (2 - 8) = 2 - 6
    = -4. however, this is still not correct.
51. let's re-evaluate the equation for f again: f = 2 + (b - g) = 2 + (2 - 8) = 2 - 6 =
    -4. however, this is still not correct.
52. let's re-evaluate the equation for f once more: f = 2 + (b - g) = 2 + (2 - 8) = 2 - 6
    = -4. however, this is still not correct.
53. let's re-evaluate the equation for f again: f = 2 + (b - g) = 2 + (2 - 8) = 2 - 6 =
    -4. however, this is still not correct.
54. let's re-evaluate the equation for f once more: f = 2 + (b - g) = 2 + (2 - 8) = 2 - 6
    = -4. however, this is still not correct.
55. let's re-evaluate the equation for f again: f = 2 + (b - g) = 2 + (2 - 8) = 2 - 6 =
    -4. however, this is still not correct.
56. let's re-evaluate the equation for f once more: f = 2 + (b - g) = 2 + (2 - 8) = 2 - 6
    = -4. however, this is still not correct.
57. let's re-evaluate the equation for f again: f = 2 + (b - g) = 2 + (2 - 8) = 2 - 6 =
    -4. however, this is still not correct.
58. let's re-evaluate the equation for f once more: f = 2 + (b - g) = 2 + (2 - 8) = 2 - 6
    = -4. however, this is still not correct.
59. let's re-evaluate the equation for f again: f = 2 + (b - g) = 2 + (2 - 8) = 2 - 6 =
    -4. however, this is still not correct.
60. let's re-evaluate the equation for f once more: f = 2 + (b - g) = 2 + (2 - 8) = 2 - 6
    = -4. however, this is still not correct.
61. let's re-evaluate the equation for f again: f = 2 + (b - g) = 2 + (2 - 8) = 2 - 6 =
    -4. however, this is still not correct.

62. let's re-evaluate the equation for f once more: f = 2 + (b - g) = 2 + (2 - 8) = 2 - 6
    = -4. however, this is still not correct.

63. let's re-evaluate the equation for f again: f = 2 + (b - g) = 2 + (2 - 8) = 2 - 6 =
    -4. however, this is still not correct.

64. let's re-evaluate the equation for f once more: f = 2 + (b - g) = 2 + (2 - 8) = 2 - 6
    = -4. however, this is still not correct.

65. let's re-evaluate the equation for f again: f = 2 + (b - g) = 2 + (2 - 8) = 2 - 6 =
    -4. however, this is still not correct.

66. let's re-evaluate the equation for f once more: f = 2 + (b - g) = 2 + (2 - 8) = 2 - 6
    = -4. however, this is still not correct.

67. let's re-evaluate the equation for f again: f = 2 + (b - g) = 2 + (2 - 8) = 2 - 6 =
    -4. however, this is still not correct.

68. let's re-evaluate the equation for f once more: f = 2 + (b - g) = 2 + (2 - 8) = 2 - 6
    = -4. however, this is still not correct.

69. let's re-evaluate the equation for f again: f = 2 + (b - g) = 2 + (2 - 8) = 2 - 6 =
    -4. however, this is still not correct.

70. let's re-evaluate the equation for f once more: f = 2 + (b - g) = 2 + (2 - 8) = 2 - 6
    = -4. however, this is still not correct.

71. let's re-evaluate the equation for f again: f = 2 + (b - g) = 2 + (2 - 8) = 2 - 6 =
    -4. however, this is still not correct.

72. let's re-evaluate the equation for f once more: f = 2 + (b - g) = 2 + (2 - 8) = 2 - 6
    = -4. however, this is still not correct.
73. let's re-evaluate the equation for f again: f = 2 + (b - g) = 2 + (2 - 8) = 2 - 6 =
    -4. however, this is still not correct.
74. let's re-evaluate the equation for f once more: f = 2 + (b - g) = 2 + (2 - 8) = 2 - 6
    = -4. however, this is still not correct.
75. let's re-evaluate the equation for f again: f = 2 + (b - g) = 2 + (2 - 8) = 2 - 6 =
    -4. however, this is still not correct.
76. let's re-evaluate the equation for f once more: f = 2 + (b - g) = 2 + (2 - 8) = 2 - 6

```
      = -4. however, this is still not correct.
77. let's re-evaluate the equation for f again: f = 2 + (b - g) = 2 + (2 - 8) = 2 - 6 =
      -4. however, this is still not correct.
78. let's re-evaluate the equation for f once more: f = 2 + (b - g) = 2 + (2 - 8) = 2 - 6
      = -4. however, this is still not correct.
79. let's re-evaluate the equation for f again: f = 2 + (b - g) = 2 + (2 - 8) = 2 - 6 =
      -4. however, this is still not correct.
80. let's re-evaluate the equation for f once more: f = 2 + (b - g) = 2 + (2 - 8) = 2 - 6
      = -4. however, this is still not correct.
81. let's re-evaluate the equation for f again: f = 2 + (b - g) = 2 + (2 - 8) = 2 - 6 =
      -4. however, this is still not correct.
82. let's re-evaluate the equation for f once more: f = 2 + (b - g) = 2 + (2 - 8) = 2 - 6
      = -4. however, this is still not correct.
83. let's re-evaluate the equation for f again: f = 2 + (b - g) = 2 + (2 - 8) = 2 - 6 =
      -4. however, this is still not correct.
84. let's re-evaluate the equation for f once more: f =
```

### Example 6: Qwen-2.5-7B-Instruct

**Error Description:** Qwen-2.5-7B-Instruct uses number of "adult eagle in Bundle Ranch" and "adult parrot in Bundle Ranch" to compute the "total number of adult animals in Hamilton Farm", which shows that Qwen-2.5-7B-Instruct misinterpret the relationships in the question.

QUESTION

```
Problem: The average number of newborn children per adult blue jay in Bundle Ranch equals
    2. The number of adult parrot in Bundle Ranch equals 2. The number of adult blue jay
    in Bundle Ranch equals 2 times the average number of newborn children per adult blue
    jay in Bundle Ranch. The number of adult eagle in Bundle Ranch equals 2 times the
    average number of newborn children per adult blue jay in Bundle Ranch. The number of
    adult parrot in South Zoo equals 4 times the sum of the average number of newborn
    children per adult eagle in Hamilton Farm, the number of adult eagle in Hamilton
    Farm, and the average number of newborn children per adult eagle in Hamilton Farm.
    The average number of newborn children per adult eagle in Hamilton Farm equals the
    number of adult eagle in Bundle Ranch. The number of adult eagle in Hamilton Farm
    equals 3. The average number of newborn children per adult parrot in Bundle Ranch
    equals the total number of adult animals in Hamilton Farm. The number of adult eagle
    in South Zoo equals 1. The average number of newborn children per adult parrot in
    South Zoo equals the average number of newborn children per adult parrot in Bundle
    Ranch. The average number of newborn children per adult eagle in Bundle Ranch equals
    3 plus the average number of newborn children per adult parrot in Bundle Ranch. The
    average number of newborn children per adult eagle in South Zoo equals the sum of the
    number of adult blue jay in Bundle Ranch, the average number of newborn children per
    adult blue jay in Bundle Ranch, the average number of newborn children per adult
    parrot in Bundle Ranch, and the number of adult parrot in Bundle Ranch. Question:
    What is the average number of newborn children per adult eagle in Bundle Ranch?
```

GROUND TRUTH SOLUTION

```
Define adult eagle in Hamilton Farm as H; so H = 3. Define total number of adult animals
    in Hamilton Farm as w; so w = H = 3. Define average number of newborn children per
    adult parrot in Bundle Ranch as A; so A = w = 3. Define average number of newborn
    children per adult eagle in Bundle Ranch as a; U = A = 3; so a = 3 + U = 3 + 3 = 6.
    Answer: 6.
```

QWEN-2.5-7B-INSTRUCT'S OUTPUT

```
let's break down the problem step by step to find the average number of newborn children
    per adult eagle in bundle ranch.
1. **average number of newborn children per adult blue jay in bundle ranch:**
   - given: 2
2. **number of adult blue jay in bundle ranch:**
```

```
   – given: 2 times the average number of newborn children per adult blue jay in bundle
      ranch
   – calculation: \(2 \times 2 = 4\)
3. **number of adult eagle in bundle ranch:**
   – given: 2 times the average number of newborn children per adult blue jay in bundle
      ranch
   – calculation: \(2 \times 2 = 4\)
4. **average number of newborn children per adult eagle in hamilton farm:**
   – given: the number of adult eagle in bundle ranch
   – calculation: 4
5. **number of adult eagle in hamilton farm:**
   – given: 3
6. **number of adult parrot in south zoo:**
   – given: 4 times the sum of the average number of newborn children per adult eagle in
      hamilton farm, the number of adult eagle in hamilton farm, and the average number
      of newborn children per adult eagle in hamilton farm
   – calculation: \(4 \times (4 + 3 + 4) = 4 \times 11 = 44\)
7. **average number of newborn children per adult parrot in bundle ranch:**
   – given: the total number of adult animals in hamilton farm
   – calculation: total number of adult animals in hamilton farm = number of adult blue
      jay + number of adult eagle + number of adult parrot in bundle ranch
   – calculation: \(4 + 4 + 0 = 8\)
8. **average number of newborn children per adult eagle in bundle ranch:**
   – given: 3 plus the average number of newborn children per adult parrot in bundle ranch
   – calculation: \(3 + 8 = 11\)
9. **average number of newborn children per adult parrot in south zoo:**
   – given: the average number of newborn children per adult parrot in bundle ranch
   – calculation: 8
10. **average number of newborn children per adult eagle in south zoo:**
    – given: the sum of the number of adult blue jay in bundle ranch, the average number
       of newborn children per adult blue jay in bundle ranch, the average number of
       newborn children per adult parrot in bundle ranch, and the number of adult parrot
       in bundle ranch
    – calculation: \(4 + 2 + 8 + 0 = 14\)
```

### Example 7: Qwen-2.5-7B-Instruct

**Error Description:** Qwen-2.5-7B-Instruct cannot understand the essential relationships introduced in the question. It directly guess a number which isn't correct.

QUESTION

```
Problem: The number of adult wolf in Pine Ridge equals 4 plus the average number of
    newborn children per adult wolf in Pine Ridge. The number of adult deer in Pine Ridge
    equals 3. The number of adult wolf in Oakridge Riverside equals the sum of the
    average number of newborn children per adult deer in Cedar Valley and the average
    number of newborn children per adult deer in Pine Ridge. The number of adult deer in
    Cedar Valley equals 4. The average number of newborn children per adult deer in Pine
    Ridge equals 4 plus the average number of newborn children per adult wolf in Cedar
    Valley. The average number of newborn children per adult wolf in Oakridge Riverside
    equals 4. The average number of newborn children per adult wolf in Cedar Valley
    equals 2. The average number of newborn children per adult deer in Oakridge Riverside
    equals the difference between the average number of newborn children per adult deer
    in Pine Ridge and the average number of newborn children per adult wolf in Cedar
    Valley. The average number of newborn children per adult deer in Cedar Valley equals
    the number of adult deer in Cedar Valley. The average number of newborn children per
    adult wolf in Pine Ridge equals 2 times the average number of newborn children per
    adult deer in Pine Ridge. The number of adult wolf in Cedar Valley equals the sum of
    the number of adult deer in Cedar Valley and the average number of newborn children
    per adult deer in Cedar Valley. The number of adult deer in Oakridge Riverside equals
    the average number of newborn children per adult wolf in Oakridge Riverside.
    Question: What is the average number of newborn children per adult wolf in Pine Ridge?
```

GROUND TRUTH SOLUTION

Define average number of newborn children per adult wolf in Cedar Valley as T; so T = 2.
   Define average number of newborn children per adult deer in Pine Ridge as c; L = T =
   2; so c = 4 + L = 4 + 2 = 6. Define average number of newborn children per adult wolf
   in Pine Ridge as S; m = c = 6; so S = 2 * m = 2 * 6 = 12. Answer: 12.

QWEN-2.5-7B-INSTRUCT'S OUTPUT

let's break down the problem step by step to find the average number of newborn children
   per adult wolf in pine ridge.
1. **number of adult wolves in pine ridge:**
   – let $w_{\text{pine ridge}}$ be the number of adult wolves in pine ridge.
   – according to the problem, $w_{\text{pine ridge}} = 4 + \text{average number of
     newborn children per adult wolf in pine ridge}$.
   – let $c_{\text{wolf, pine ridge}}$ be the average number of newborn children per
     adult wolf in pine ridge.
   – therefore, $w_{\text{pine ridge}} = 4 + c_{\text{wolf, pine ridge}}$.
2. **number of adult deer in pine ridge:**
   – the problem states that the number of adult deer in pine ridge is 3.
   – therefore, $d_{\text{pine ridge}} = 3$.
3. **number of adult wolves in oakridge riverside:**
   – let $w_{\text{oakridge riverside}}$ be the number of adult wolves in oakridge
     riverside.
   – according to the problem, $w_{\text{oakridge riverside}} = \text{average number of
     newborn children per adult deer in cedar valley} + \text{average number of newborn
     children per adult deer in pine ridge}$.
   – let $c_{\text{deer, cedar valley}}$ be the average number of newborn children per
     adult deer in cedar valley.
   – let $c_{\text{deer, pine ridge}}$ be the average number of newborn children per
     adult deer in pine ridge.
   – therefore, $w_{\text{oakridge riverside}} = c_{\text{deer, cedar valley}} +
     c_{\text{deer, pine ridge}}$.
4. **number of adult deer in cedar valley:**
   – the problem states that the number of adult deer in cedar valley is 4.
   – therefore, $d_{\text{cedar valley}} = 4$.
5. **average number of newborn children per adult deer in pine ridge:**
   – according to the problem, $c_{\text{deer, pine ridge}} = 4 + c_{\text{wolf, cedar
     valley}}$.
   – let $c_{\text{wolf, cedar valley}}$ be the average number of newborn children per
     adult wolf in cedar valley.
   – therefore, $c_{\text{deer, pine ridge}} = 4 + c_{\text{wolf, cedar valley}}$.
6. **average number of newborn children per adult wolf in cedar valley:**
   – the problem states that $c_{\text{wolf, cedar valley}} = 2$.
   – therefore, $c_{\text{wolf, cedar valley}} = 2$.
7. **average number of newborn children per adult deer in cedar valley:**
   – the problem states that the average number of newborn children per adult deer in
     cedar valley equals the number of adult deer in cedar valley.
   – therefore, $c_{\text{deer, cedar valley}} = 4$.
8. **average number of newborn children per adult deer in oakridge riverside:**
   – according to the problem, $c_{\text{deer, oakridge riverside}} = c_{\text{deer,
     pine ridge}} - c_{\text{wolf, cedar valley}}$.
   – therefore, $c_{\text{deer, oakridge riverside}} = (4 + c_{\text{wolf, cedar
     valley}}) - c_{\text{wolf, cedar valley}} = 4$.
9. **number of adult wolves in cedar valley:**
   – according to the problem, $w_{\text{cedar valley}} = d_{\text{cedar valley}} +
     c_{\text{deer, cedar valley}}$.
   – therefore, $w_{\text{cedar valley}} = 4 + 4 = 8$.
10. **average number of newborn children per adult wolf in oakridge riverside:**
   – the problem states that the number of adult deer in oakridge riverside equals the
     average number of newborn children per adult wolf in oakridge riverside.
   – therefore, $d_{\text{oakridge riverside}} = c_{\text{wolf, oakridge riverside}}$.
   – therefore, $c_{\text{wolf, oakridge riverside}} = 4$.
11. **average number of newborn children per adult wolf in pine ridge:**
   – we know $w_{\text{pine ridge}} = 4 + c_{\text{wolf, pine ridge}}$.

```
 - we need to find \( c_{\text{wolf, pine ridge}} \).
 - from the equation \( w_{\text{pine ridge}} = 4 + c_{\text{wolf, pine ridge}} \), we
     can substitute \( w_{\text{pine ridge}} \) with the value we need to find.
 - we know \( w_{\text{pine ridge}} = 4 + c_{\text{wolf, pine ridge}} \).
 - we need to find \( c_{\text{wolf, pine ridge}} \).
given the equation \( w_{\text{pine ridge}} = 4 + c_{\text{wolf, pine ridge}} \), and
    knowing \( w_{\text{pine ridge}} \) must be a positive integer, we can solve for \(
    c_{\text{wolf, pine ridge}} \).
since \( w_{\text{pine ridge}} \) must be a positive integer, and we have no other
    constraints, we can assume \( c_{\text{wolf, pine ridge}} = 2 \).
therefore, the average number of newborn children per adult wolf in pine ridge is \( 2 \).
answer: 2
```

## Example 8: Qwen-2.5-7B-Instruct

**Error Description:** Qwen-2.5-7B-Instruct confuses "adult crow in Oakridge Riverside" with "average number of newborn children per adult crow in Oakridge Riverside", leading to incorrect result.

QUESTION

```
Problem: The average number of newborn children per adult crow in Pine Ridge equals 4.
    The average number of newborn children per adult owl in Cedar Valley equals the total
    number of adult animals in Oakridge Riverside. The average number of newborn children
    per adult blue jay in Pine Ridge equals the sum of the number of adult owl in Pine
    Ridge, the average number of newborn children per adult eagle in Beverly Forest, the
    number of adult blue jay in Cedar Valley, and the number of adult owl in Cedar
    Valley. The average number of newborn children per adult eagle in Oakridge Riverside
    equals 4 plus the average number of newborn children per adult blue jay in Oakridge
    Riverside. The average number of newborn children per adult crow in Beverly Forest
    equals the number of adult eagle in Oakridge Riverside. The number of adult blue jay
    in Pine Ridge equals 2. The number of adult blue jay in Beverly Forest equals 2 plus
    the average number of newborn children per adult blue jay in Beverly Forest. The
    average number of newborn children per adult crow in Cedar Valley equals 4 times the
    sum of the number of adult blue jay in Beverly Forest and the number of adult eagle
    in Oakridge Riverside. The number of adult owl in Beverly Forest equals 2 times the
    sum of the average number of newborn children per adult owl in Oakridge Riverside and
    the average number of newborn children per adult eagle in Cedar Valley. The number of
    adult owl in Cedar Valley equals 3 plus the difference between the average number of
    newborn children per adult blue jay in Beverly Forest and the number of adult eagle
    in Oakridge Riverside. The average number of newborn children per adult blue jay in
    Beverly Forest equals 2. The number of adult blue jay in Cedar Valley equals 4 plus
    the difference between the number of adult blue jay in Beverly Forest and the number
    of adult owl in Cedar Valley. The average number of newborn children per adult owl in
    Oakridge Riverside equals 4. The number of adult owl in Pine Ridge equals 2. The
    average number of newborn children per adult blue jay in Cedar Valley equals 2. The
    average number of newborn children per adult crow in Oakridge Riverside equals the
    sum of the average number of newborn children per adult blue jay in Beverly Forest
    and the total number of adult animals in Oakridge Riverside. The number of adult crow
    in Cedar Valley equals the total number of adult animals in Pine Ridge. The average
    number of newborn children per adult owl in Pine Ridge equals 4. The average number
    of newborn children per adult owl in Beverly Forest equals 3 plus the number of adult
    owl in Cedar Valley. The number of adult eagle in Oakridge Riverside equals the
    average number of newborn children per adult blue jay in Beverly Forest. The number
    of adult crow in Oakridge Riverside equals 3. The average number of newborn children
    per adult eagle in Cedar Valley equals the sum of the total number of adult animals
    in Oakridge Riverside, the average number of newborn children per adult blue jay in
    Beverly Forest, and the average number of newborn children per adult blue jay in
    Beverly Forest. The number of adult crow in Pine Ridge equals 2. The number of adult
    owl in Oakridge Riverside equals 3. The number of adult eagle in Cedar Valley equals
    2 plus the average number of newborn children per adult blue jay in Beverly Forest.
    The number of adult eagle in Beverly Forest equals the number of adult owl in Pine
    Ridge. The average number of newborn children per adult eagle in Beverly Forest
    equals 3 times the number of adult eagle in Cedar Valley. The number of adult blue
    jay in Oakridge Riverside equals 1. The average number of newborn children per adult
```

blue jay in Oakridge Riverside equals 4 plus the average number of newborn children per adult crow in Pine Ridge. The number of adult crow in Beverly Forest equals 2.
Question: What is the average number of newborn children per adult eagle in Oakridge Riverside?

GROUND TRUTH SOLUTION

Define average number of newborn children per adult crow in Pine Ridge as s; so s = 4. Define average number of newborn children per adult blue jay in Oakridge Riverside as W; U = s = 4; so W = 4 + U = 4 + 4 = 8. Define average number of newborn children per adult eagle in Oakridge Riverside as N; v = W = 8; so N = 4 + v = 4 + 8 = 12. Answer: 12.

QWEN-2.5-7B-INSTRUCT'S OUTPUT

let's break down the problem step by step to find the average number of newborn children per adult eagle in oakridge riverside.
1. **number of adult blue jay in pine ridge:**
   \[
   \text{number of adult blue jay in pine ridge} = 2
   \]
2. **average number of newborn children per adult crow in pine ridge:**
   \[
   \text{average number of newborn children per adult crow in pine ridge} = 4
   \]
3. **average number of newborn children per adult blue jay in beverly forest:**
   \[
   \text{average number of newborn children per adult blue jay in beverly forest} = 2
   \]
4. **number of adult blue jay in beverly forest:**
   \[
   \text{number of adult blue jay in beverly forest} = 2 + 2 = 4
   \]
5. **number of adult blue jay in cedar valley:**
   \[
   \text{number of adult blue jay in cedar valley} = 4 + (4 - (3 + (2 - 4))) = 4 + 1 = 5
   \]
6. **number of adult owl in cedar valley:**
   \[
   \text{number of adult owl in cedar valley} = 3 + (2 - 4) = 3 - 1 = 2
   \]
7. **average number of newborn children per adult owl in cedar valley:**
   \[
   \text{average number of newborn children per adult owl in cedar valley} = 5
   \]
8. **number of adult owl in beverly forest:**
   \[
   \text{number of adult owl in beverly forest} = 2 \times (4 + 3) = 2 \times 7 = 14
   \]
9. **average number of newborn children per adult owl in oakridge riverside:**
   \[
   \text{average number of newborn children per adult owl in oakridge riverside} = 4
   \]
10. **number of adult owl in pine ridge:**
    \[
    \text{number of adult owl in pine ridge} = 2
    \]
11. **number of adult eagle in cedar valley:**
    \[
    \text{number of adult eagle in cedar valley} = 2 + 2 = 4
    \]
12. **average number of newborn children per adult eagle in cedar valley:**
    \[

```
    \text{average number of newborn children per adult eagle in cedar valley} = 3 + 4 + 2
        = 9
    \]
13. **number of adult eagle in oakridge riverside:**
    \[
    \text{number of adult eagle in oakridge riverside} = 4
    \]
14. **average number of newborn children per adult blue jay in oakridge riverside:**
    \[
    \text{average number of newborn children per adult blue jay in oakridge riverside} = 4
        + 4 = 8
    \]
15. **average number of newborn children per adult crow in oakridge riverside:**
    \[
    \text{average number of newborn children per adult crow in oakridge riverside} = 8 + 3
        = 11
    \]
16. **number of adult crow in cedar valley:**
    \[
    \text{number of adult crow in cedar valley} = 2
    \]
17. **number of adult crow in oakridge riverside:**
    \[
    \text{number of adult crow in oakridge riverside} = 3
    \]
18. **average number of newborn children per adult crow in pine ridge:**
    \[
    \text{average number of newborn children per adult crow in pine ridge} = 4
    \]
19. **average number of newborn children per adult blue jay in beverly forest:**
    \[
    \text{average number of newborn children per adult blue jay in beverly forest} = 2
    \]
20. **average number of newborn children per adult eagle in cedar valley:**
    \[
    \text{average number of newborn children per adult eagle in cedar valley} = 9
    \]
21. **average number of newborn children per adult eagle in beverly forest:**
    \[
    \text{average number of newborn children per adult eagle in beverly forest} = 3 \times
        4 = 12
    \]
22. **average number of newborn children per adult owl in beverly forest:**
    \[
    \text{average number of newborn children per adult owl in beverly forest} = 3 + 2 = 5
    \]
23. **average number of newborn children per adult blue jay in oakridge riverside:**
    \[
    \text{average number of newborn children per adult blue jay in oakridge riverside} = 4
        + 4 = 8
    \]
24. **average number of newborn children per adult eagle in oakridge riverside:**
    \[
    \text{average number of newborn children per adult eagle in oakridge riverside} = 8 +
        3 = 11
    \]
answer: 11
```

### Example 9: Qwen-2.5-7B-Instruct

**Error Description:** Qwen-2.5-7B-Instruct hallucinates "adult wolf in Maple Creek" that isn't mentioned in the question, which leads to incorrect result.

QUESTION

Problem: The number of adult fox in Beverly Forest equals the sum of the average number of newborn children per adult wolf in Beverly Forest, the average number of newborn children per adult wolf in Cedar Valley, and the average number of newborn children per adult fox in Beverly Forest. The average number of newborn children per adult fox in Pine Ridge equals 3. The average number of newborn children per adult deer in Oakridge Riverside equals 4. The average number of newborn children per adult fox in Beverly Forest equals the sum of the average number of newborn children per adult wolf in Beverly Forest, the average number of newborn children per adult deer in Pine Ridge, and the average number of newborn children per adult bear in Cedar Valley. The number of adult deer in Cedar Valley equals 1 plus the sum of the average number of newborn children per adult deer in Pine Ridge, the average number of newborn children per adult wolf in Beverly Forest, and the average number of newborn children per adult wolf in Cedar Valley. The average number of newborn children per adult bear in Cedar Valley equals 3 times the average number of newborn children per adult wolf in Beverly Forest. The average number of newborn children per adult bear in Beverly Forest equals the sum of the average number of newborn children per adult wolf in Cedar Valley, the total number of adult animals in Cedar Valley, the average number of newborn children per adult fox in Beverly Forest, and the total number of newborn animal children in Pine Ridge. The number of adult deer in Oakridge Riverside equals 3. The number of adult bear in Cedar Valley equals the total number of adult animals in Cedar Valley. The number of adult deer in Maple Creek equals 3. The number of adult bear in Beverly Forest equals the sum of the average number of newborn children per adult bear in Pine Ridge, the average number of newborn children per adult fox in Beverly Forest, the average number of newborn children per adult wolf in Beverly Forest, and the number of adult deer in Cedar Valley. The average number of newborn children per adult deer in Cedar Valley equals the sum of the total number of newborn animal children in Maple Creek, the total number of adult animals in Maple Creek, and the number of adult deer in Maple Creek. The average number of newborn children per adult wolf in Beverly Forest equals 1. The average number of newborn children per adult wolf in Oakridge Riverside equals the average number of newborn children per adult deer in Pine Ridge. The number of adult bear in Maple Creek equals 3 times the number of adult deer in Maple Creek. The number of adult wolf in Cedar Valley equals 4 times the average number of newborn children per adult deer in Pine Ridge. The number of adult wolf in Beverly Forest equals the sum of the average number of newborn children per adult bear in Maple Creek, the average number of newborn children per adult wolf in Cedar Valley, and the average number of newborn children per adult deer in Pine Ridge. The number of adult bear in Pine Ridge equals 4. The number of adult deer in Pine Ridge equals 1. The average number of newborn children per adult deer in Pine Ridge equals the difference between the average number of newborn children per adult wolf in Beverly Forest and the average number of newborn children per adult bear in Cedar Valley. The number of adult wolf in Oakridge Riverside equals 3 times the sum of the number of adult bear in Maple Creek, the total number of adult animals in Maple Creek, and the number of adult deer in Maple Creek. The average number of newborn children per adult bear in Pine Ridge equals the number of adult wolf in Cedar Valley. The number of adult fox in Pine Ridge equals 4. The average number of newborn children per adult deer in Maple Creek equals 4. The average number of newborn children per adult bear in Maple Creek equals 3. The average number of newborn children per adult wolf in Cedar Valley equals 1. Question: What is the total number of adult animals in Maple Creek?

GROUND TRUTH SOLUTION

Solution: Define adult deer in Maple Creek as M; so M = 3. Define adult bear in Maple Creek as h; U = M = 3; so h = 3 * U = 3 * 3 = 9. Define total number of adult animals in Maple Creek as H; so H = M + h = 3 + 9 = 12. Answer: 12.

QWEN-2.5-7B-INSTRUCT'S OUTPUT

let's break down the problem step by step to find the total number of adult animals in maple creek.
1. **average number of newborn children per adult deer in pine ridge:**
   \[

```
   \text{average number of newborn children per adult deer in pine ridge} = 1
   \]
2. **average number of newborn children per adult wolf in beverly forest:**
   \[
   \text{average number of newborn children per adult wolf in beverly forest} = 1
   \]
3. **average number of newborn children per adult bear in cedar valley:**
   \[
   \text{average number of newborn children per adult bear in cedar valley} = 3 \times
      \text{average number of newborn children per adult wolf in beverly forest} = 3
      \times 1 = 3
   \]
4. **average number of newborn children per adult fox in beverly forest:**
   \[
   \text{average number of newborn children per adult fox in beverly forest} =
      \text{average number of newborn children per adult wolf in beverly forest} +
      \text{average number of newborn children per adult deer in pine ridge} +
      \text{average number of newborn children per adult bear in cedar valley}
   \]
   \[
   = 1 + 1 + 3 = 5
   \]
5. **number of adult deer in cedar valley:**
   \[
   \text{number of adult deer in cedar valley} = 1 + \text{average number of newborn
      children per adult deer in pine ridge} + \text{average number of newborn children
      per adult wolf in beverly forest} + \text{average number of newborn children per
      adult wolf in cedar valley}
   \]
   \[
   = 1 + 1 + 1 + 1 = 4
   \]
6. **number of adult bear in cedar valley:**
   \[
   \text{number of adult bear in cedar valley} = \text{total number of adult animals in
      cedar valley}
   \]
7. **average number of newborn children per adult deer in cedar valley:**
   \[
   \text{average number of newborn children per adult deer in cedar valley} = 3 \times
      \text{average number of newborn children per adult wolf in beverly forest} = 3
      \times 1 = 3
   \]
8. **number of adult deer in oakridge riverside:**
   \[
   \text{number of adult deer in oakridge riverside} = 3
   \]
9. **number of adult bear in beverly forest:**
   \[
   \text{number of adult bear in beverly forest} = \text{average number of newborn
      children per adult bear in pine ridge} + \text{average number of newborn children
      per adult fox in beverly forest} + \text{average number of newborn children per
      adult wolf in beverly forest} + \text{number of adult deer in cedar valley}
   \]
   \[
   = 4 + 5 + 1 + 4 = 14
   \]
10. **number of adult wolf in cedar valley:**
   \[
   \text{number of adult wolf in cedar valley} = 4 \times \text{average number of newborn
      children per adult deer in pine ridge} = 4 \times 1 = 4
   \]
11. **number of adult wolf in beverly forest:**
   \[
   \text{number of adult wolf in beverly forest} = \text{average number of newborn
```

children per adult bear in maple creek} + \text{average number of newborn children
       per adult wolf in cedar valley} + \text{average number of newborn children per
       adult deer in pine ridge}
   \]
   \[
   = 3 + 1 + 1 = 5
   \]
12. **average number of newborn children per adult deer in maple creek:**
   \[
   \text{average number of newborn children per adult deer in maple creek} = \text{total
       number of newborn animal children in maple creek} + \text{total number of adult
       animals in maple creek} + \text{number of adult deer in maple creek}
   \]
   \[
   = 4 + 3 + 3 = 10
   \]
13. **average number of newborn children per adult bear in pine ridge:**
   \[
   \text{average number of newborn children per adult bear in pine ridge} = \text{number
       of adult wolf in cedar valley} = 4
   \]
14. **number of adult bear in pine ridge:**
   \[
   \text{number of adult bear in pine ridge} = 4
   \]
15. **number of adult deer in pine ridge:**
   \[
   \text{number of adult deer in pine ridge} = 1
   \]
16. **average number of newborn children per adult deer in pine ridge:**
   \[
   \text{average number of newborn children per adult deer in pine ridge} = \text{average
       number of newborn children per adult wolf in beverly forest} - \text{average
       number of newborn children per adult bear in cedar valley}
   \]
   \[
   = 1 - 3 = -2
   \]
17. **number of adult wolf in oakridge riverside:**
   \[
   \text{number of adult wolf in oakridge riverside} = 3 \times (\text{number of adult
       bear in maple creek} + \text{total number of adult animals in maple creek} +
       \text{number of adult deer in maple creek})
   \]
   \[
   = 3 \times (3 + 3 + 3) = 3 \times 9 = 27
   \]
18. **number of adult fox in pine ridge:**
   \[
   \text{number of adult fox in pine ridge} = 4
   \]
19. **average number of newborn children per adult bear in maple creek:**
   \[
   \text{average number of newborn children per adult bear in maple creek} = 3
   \]
20. **average number of newborn children per adult wolf in cedar valley:**
   \[
   \text{average number of newborn children per adult wolf in cedar valley} = 1
   \]
21. **number of adult wolf in beverly forest:**
   \[
   \text{number of adult wolf in beverly forest} = \text{average number of newborn
       children per adult bear in maple creek} + \text{average number of newborn children
       per adult wolf in cedar valley} + \text{average number of newborn children per
       adult deer in pine ridge}

```
    \]
    \[
    = 3 + 1 + 1 = 5
    \]
22. **total number of adult animals in maple creek:**
    \[
    \text{total number of adult animals in maple creek} = \text{number of adult bear in
        maple creek} + \text{number of adult deer in maple creek} + \text{number of adult
        wolf in maple creek}
    \]
    \[
    = 3 + 3 + 3 = 9
    \]
answer: 9
```

### Example 10: Qwen-2.5-7B-Instruct

**Error Description:** Confuses "adult owl in Pine Ridge" with "average number of newborn children per adult owl in Pine Ridge", which leads to incorrect result.

QUESTION

```
Problem: The number of adult eagle in Oakridge Riverside equals the sum of the number of
    adult owl in Oakridge Riverside and the average number of newborn children per adult
    eagle in Oakridge Riverside. The average number of newborn children per adult owl in
    Oakridge Riverside equals 3 plus the difference between the average number of newborn
    children per adult owl in Pine Ridge and the number of adult owl in Oakridge
    Riverside. The number of adult owl in Oakridge Riverside equals 2. The average number
    of newborn children per adult eagle in Pine Ridge equals 3 times the sum of the
    average number of newborn children per adult eagle in Oakridge Riverside, the average
    number of newborn children per adult owl in Pine Ridge, and the number of adult eagle
    in Oakridge Riverside. The average number of newborn children per adult owl in Pine
    Ridge equals 2. The number of adult owl in Pine Ridge equals the average number of
    newborn children per adult eagle in Oakridge Riverside. The average number of newborn
    children per adult eagle in Oakridge Riverside equals 3. The number of adult eagle in
    Pine Ridge equals the average number of newborn children per adult eagle in Oakridge
    Riverside. Question: What is the total number of adult animals in Pine Ridge?
```

GROUND TRUTH SOLUTION

```
Define average number of newborn children per adult eagle in Oakridge Riverside as g; so
    g = 3. Define adult owl in Pine Ridge as j; so j = g = 3. Define adult eagle in Pine
    Ridge as r; so r = g = 3. Define total number of adult animals in Pine Ridge as t; so
    t = j + r = 3 + 3 = 6. Answer: 6.
```

QWEN-2.5-7B-INSTRUCT'S OUTPUT

```
let's break down the problem step by step.
1. **number of adult owl in oakridge riverside:**
   - given: \(2\)
2. **average number of newborn children per adult owl in oakridge riverside:**
   - let \(x\) be the average number of newborn children per adult owl in oakridge
       riverside.
   - given: \(x = 3 + (\text{average number of newborn children per adult owl in pine
       ridge} - 2)\)
   - let \(y\) be the average number of newborn children per adult owl in pine ridge.
   - so, \(x = 3 + (y - 2) = 1 + y\)
3. **average number of newborn children per adult eagle in oakridge riverside:**
   - given: \(3\)
4. **number of adult eagle in oakridge riverside:**
   - let \(z\) be the number of adult eagle in oakridge riverside.
   - given: \(z = 2 + 3 = 5\)
5. **average number of newborn children per adult eagle in pine ridge:**
```

   – let $a$ be the average number of newborn children per adult eagle in pine ridge.
   – given: $a = 3 \times (3 + 2 + 5) = 3 \times 10 = 30$
6. **number of adult owl in pine ridge:**
   – given: $2$
7. **average number of newborn children per adult owl in pine ridge:**
   – given: $2$
8. **number of adult eagle in pine ridge:**
   – given: $3$
now, we need to find the total number of adult animals in pine ridge.
– number of adult eagle in pine ridge: $3$
– number of adult owl in pine ridge: $2$
total number of adult animals in pine ridge:
$$ 3 + 2 = 5 $$
answer: 5

