# OpenReview forum: "GSM-$\infty$: How Do your LLMs Behave over Infinitely Increasing Reasoning Complexity and Context Length?"
_ICML.cc/2025/Conference — ICML 2025 poster_

### Official Review · Reviewer_GB4k · 2025-03-11

**Overall Recommendation:** 3

**Summary:**

This paper introduces a new benchmark for testing the performance of LLMs on long-context reasoning.In particular, the authors start from the abstract computational graphs of the GSK-8K problems and develop the GSM-Infinite benchmark.With the constructed benchmark, the authors observe a consistent sigmoidal decline in reasoning performance as the complexity increases.

**Claims And Evidence:**

This paper argues that a long-context reasoning benchmark should offer controllable and scalable complexity, natural noise, and support infinite data generation.The first two points are somewhat realized in GSM-Infinite, but the third point is less clear. The authors have only expanded based on GSM-8K as a seed, and it is not evident whether the proposed method can be applied to various types of mathematical benchmarks.

**Essential References Not Discussed:**

I did not notice the issue of missing references.

**Experimental Designs Or Analyses:**

The experimental designs based on the GSM-infinite are sound. I also checked the experimental details in the appendix,and I did not notice any obvious flaws.

**Methods And Evaluation Criteria:**

The methods and evaluation criteria are solid. I did not find any obvious flaws in the method and experiment sections.

**Other Comments Or Suggestions:**

It appears that the construction of GSM-Infinite is based on computational graphs, which can be applied to a broader range of mathematical and reasoning problems. Have you considered constructing multi-source datasets under this method to build“infinite”datasets, which could lead to more universal conclusions?

**Other Strengths And Weaknesses:**

This paper proposes GSM-Infinite, which synthetically constructs long-context questions based on GSM-8K as seed problems.Neither the dataset itself nor the experiments based on the dataset have significant issues. However, as the authors discussed in Appendix A.4,t here are many synthetic methods for expanding seed questions into long-context reasoning tasks with controllable quality. The pipeline designed by the authors is also rather tailored to GSM-8K, which raises concerns that the novelty and significance of this paper are limited. Therefore, I consider this paper to be at the borderline level.

**Questions For Authors:**

Have you manually checked the synthesized questions? Are there any unnatural cases? Have you conducted any statistics on the diversity and quality of the questions?

**Relation To Broader Scientific Literature:**

This paper proposes GSM-Infinite, a synthetic dataset for testing reasoning capabilities over long contexts.It serves as a complement to existing mathematical benchmarks and can test a model's reasoning abilities in longer contexts or with introduced noises.

**Theoretical Claims:**

Not applicable. This paper does not contain any theoretical claims.

---

> ### Author Rebuttal · Authors · 2025-04-01
>
> **Concern about Practicality of GSM-Infinite**
>
> We appreciate the reviewer’s concern about the practicality of GSM-Infinite, as the problem is crucial for understanding why we believe GSM-Infinite is helpful for researchers to use. We want to kindly rebut that first, different from “many synthetic methods for expanding seed questions into long-context reasoning tasks with controllable quality,” GSM-Infinite introduces two important techniques to offer important additional edges over past benchmarks.
>
> First, unlike prior work that expands GSM-8K seeds, our test examples are generated entirely from scratch using graph-based generation—no GSM-8K questions or data are used (Section 3, Pages 3–5). GSM-8K is saturated in both quantity and difficulty (Figure 5(a), Page 4), with only ~10 examples per op>8. In contrast, GSM-Infinite offers >500 test cases per op, scaling up to op=200+, saturating SOTA LLMs.
>
> Second, prior long-context reasoning benchmarks from "expanding seed" can often have noise filtered out by retrieval (RAG), making them poor tests of long-context LLMs (Figure 4(a), Page 4). Our graph-based expansion produces problems where noise is indistinguishable from essential info (Figure 4(b), Page 4), rendering RAG ineffective. GSM-Infinite scales both reasoning and context length while being fully LLM/human-free.
>
> Moreover, we believe that GSM-Infinite offers the reasoning community a new tool that evaluates LLMs solely on reasoning with complex graph traversal, without testing them with memorization of subject-specific knowledge. Recent LLMs often report metrics from MATH or AIME datasets, since prior datasets such as GSM-8K have been saturated. Although these datasets are of strong reasoning complexity, solving test examples convincingly requires a lot of math-related theorem knowledge in LLMs’ memory. We show in (Table 1, page 7) that the powerful reasoning models also see their performance decaying to zero as we gradually increase the ops. GSM-Infinite offers an alternative reasoning benchmark that is drastically less reliant on LLMs’ memory, while evaluating all LLMs based solely on their reasoning ability of traversing a complex computational graph problem with fine-grained difficulty control for the study.
>
> **Lacks to infinite dataset**
>
> Thank you for this insightful suggestion. Our current implementation, GSM-∞, focuses specifically on modeling the explicit arithmetic operations (+,-,×,÷) and implicit hierarchical relationships found in GSM-8K using computational graphs. The primary goal was to create a scalable benchmark for grade-school math reasoning with controllable complexity and context length.
>
> While the graph representation is potentially applicable to a broader range of problems, our current work does not aim to build a universal "infinite" dataset covering all reasoning types. Extending this methodology to model other complex structures, like code syntax trees or spatial relationships, is a promising direction for future research.
>
> **Dataset Quality Inspection and Overview**
>
> Questions were constantly evaluated during development via manual inspection and by feeding them to SOTA LLMs to ensure they were natural, solvable, and LLM-understandable. SOTA models demonstrate high accuracy (>88% for o3-mini on Hard op 30) on zero-noise, lower-op problems, indicating fundamental understandability.
> |op|Forward|Reverse|Score|
> |---|---|---|---|
> |10|1.00|0.96|0.98|
> |20|0.98|0.90|0.95|
> |30|0.94|0.84|0.89|
>
> Errors observed stemmed from model reasoning limitations (e.g., confusing concepts, hallucination, misinterpreting relationships), not ill-formed prompts. We collect examples in https://docs.google.com/document/d/1WP3ygB67yNUS-iliSYYjFRY4Ls81yIVcRVxpbTVTZ0o/edit?usp=sharing Part 4.
>
> Moreover, we present a table for an overview of the composition of the dataset (submitting version).
>
> | Subsets | Templates|  | | | |
> |---|---|---|---|---|---|
> | **Symbolic**        | Uniform template of variable name “V_number” |                |                        |                            |                     |
> | **Medium and Hard** | **Configuration** | **Crazy Zootopia (80%)** | **Teachers and School (10%)** |                            | **Movie Awards (10%)** |
> || Forward (50%)| 40%|5%| 5%||
> || Reverse (50%)|40%|5%|5%||
>
> The amount of variation permitted in each of our templates is different, richest for "Crazy Zootopia" and slightly more limited for the other two templates. We use an un-even partition of templates for the ensemble of our dataset. We are actively expanding the number of available templates for GSM-Infinite, now made 10 (much more than 3 when submitting), and will be released once published. **Please notice that the template proportions are fully reconfigurable, and switching to a different template won’t leads to large difference in performance.** For detailed ablation experiments of templates, please refer to (Appendix F, page 19-20).

---

### Official Review · Reviewer_6Z5g · 2025-03-14

**Overall Recommendation:** 4

**Summary:**

The paper introduces ​GSM-∞, a synthetic benchmark for evaluating long-context LLMs on grade-school math problems. It generates problems with ​controllable complexity (via computation graphs) and ​arbitrary context length (via "spider topology" noise). The paper finds that:
1. ​Sigmoid performance decay as reasoning complexity increases.
2. Reverse problems (backward reasoning) are harder than forward ones.
3. ​Exponential inference cost yields only linear performance gains.

**Claims And Evidence:**

Mostly yes.

**Essential References Not Discussed:**

N/A

**Experimental Designs Or Analyses:**

Yes.
The paper tests 18 models on zero-noise tasks and 10 on long-context.

**Methods And Evaluation Criteria:**

Yes

**Other Comments Or Suggestions:**

1. The explanation of the computation graph approach can be improved by introducing more detailed breakdowns.
2. Add more qualitative analysis of model errors.

**Other Strengths And Weaknesses:**

Strengths:
1. The paper isolates reasoning from retrieval.
2. The ablation analysis is comprehensive.

Weaknesses:
1. Lack of ablation study on different graph structures.
2. Limited Analysis when comparing models.
3. Insufficient discussion about compute costs.

**Questions For Authors:**

1. In section 3.2, how different distributions of noise impact performance? Can a carefully designed RAG system skip it?
2. How different types of graph structures impact model performance?
3. The paper finds that LLMs perform worse on reverse problems than forward ones. Whether it is due to lack of training or data in the training set. Can it be effectively improved by providing more similar data?

**Relation To Broader Scientific Literature:**

This paper is related to GSM8K, RULER, and LongBench.

**Theoretical Claims:**

N/A

---

> ### Author Rebuttal · Authors · 2025-04-01
>
> Additional figures and tables in the link (anonymous):
> https://docs.google.com/document/d/1WP3ygB67yNUS-iliSYYjFRY4Ls81yIVcRVxpbTVTZ0o/edit?usp=sharing (referred to as **Sup**)
>
> **1. No ablation on different graph structures.**
>
> Thank you. We agree studying graph structure impact is valuable. We ran Qwen 2.5 7B IT on 40k Symbolic (op=9) to analyze the relationship of graph depth and accuracy. Depth is the max node distance from the root; mean depth is the average across nodes. We categorized samples by max depth (Depth=2 to 6).
>
> Results confirm deeper graphs are harder. Accuracy decreases as depth increases:
> |Depth|2|3|4|5|6|
> |---|---|---|---|---|---|
> |Accuracy|.438|.386|.347|.317|.257|
>
> Further analysis shows accuracy decreases as *mean depth* increases. When mean depth is similar, max depth has little impact, suggesting overall complexity (mean depth) is the primary factor. A plot confirms this trend across depth settings: **Sup** Part 1.
>
> **2. Limited Analysis between models.**
> Below data from Hard subset.
> Acknowledged. Based on Table 1 (p7):
> *   **Reasoning Models:** Show significant gains over non-reasoning counterparts with identical parameters/architecture.
> |op|DeepSeek R1|DeepSeek V3|Qwen QWQ 32B Preview|Qwen 2.5 32B Instruct (non-reasoning)|
> |---|---|---|---|---|
> |10|.95|.79|.64|.61|
> |20|.94|.60|.60|.36|
> |30|.95|.36|.26|.12|
> |40|.92|.16|.13|.04|
> |AUC|8573.8|2407.9|2846.2|1399.8|
> *   **Model Size:** Within families, larger models perform better.
> |op|Qwen2.5 72B Instruct|Qwen2.5 7B Instruct|Llama 3.1 70B Instruct|Llama 3.1 8B Instruct|
> |---|---|---|---|---|
> |5|.95|.59|.73|.52|
> |10|.82|.29|.64|.40|
> |15|.76|.18|.49|.19|
> |20|.64|.05|.33|.07|
> |25|.49|.03|.26|.06|
> |30|.37|.00|.04|.02|
> |AUC|2016.375|618.500|1205.250|606.500|
> *   **Architecture:** Current hybrid models (Linear Attention/SSM) lag behind comparable Transformer models.
> |Models|Architecture|ParameterSize|AUC|
> |---|---|---|---|
> |Qwen-2.5-32B-Instruct|Transformer|32B|1405.055|
> |MiniMax-Text-01|Hybrid (Linear Attention)|456BMoE(45.9B/token)|1178.510|
> |Jamba-1.5-Large|Hybrid (SSM)|398BMoE(98B/token)|466.400|
>
> **3. No cost discussions.**
>
> We compute the cost for Llama-3.1-70B below as a reference. We evaluated 100+ examples per op across 3 subsets (use 100). We evaluated up to 40-55 ops per subset (sampling frequency decreases after op 30). Total zero-noise evaluation (~9300 examples) requires ~9.8M input / ~39.2M output tokens. Using competitive (DeepInfra) API pricing, costs are:
> |Context|Cost|
> |---|---|
> |Zero|$12.94|
> |8K|$5.83|
> |16K|$7.49|
> |32K|$12.60|
> |**Total**|**$30.43**|
>
> Larger/reasoning models cost more. With additional tables and elaboration in **Sup** Part 2., found using strides (evaluating every N ops) up to 4 has minimal impact (<3% AUC change) on results. Reducing samples per op (e.g., 100 or 50) also yields similar AUC, allowing further cost reduction.
>
> **4. Need more detailed graph generation process.**
>
> Acknowledged. We will make sure to add greater and finer step-by-step details about the generation process in the revised version of the paper.
>
> **5. No Model Error Discussion**
>
> We analyzed errors for Llama-3.1-8B and Qwen-2.5-7B on op=5 problems. Examples available in **Sup** Part 4.
> Common error types:
> |ErrorType|ExampleIndices(Internal)|
> |---|---|
> |Confuse similar concepts|(1),(8),(10)|
> |Hallucinate concepts not in question|(2),(4),(9)|
> |Distracted by irrelevant variables|(3)|
> |Misinterpret relationships|(5),(6),(7)|
>
> **6. Noise distribution impact & RAG.**
>
> Our ablation (Sec 5.3, Fig 7c/d) shows model performance is robust to noise types, but RAG struggles significantly with our "Spider Topology" noise. Our RAG baseline is strong (all-mpnet-base-v2 retriever, Llama-3.1-70B decoder, 2k token budget, strong benchmark performance). RAG fails because it uses contextual (semantic similarity) search. Spider Topology adds irrelevant nodes semantically close to and connected to core graph nodes. RAG's fixed budget means noise can displace relevant chunks. RAG retrievers evaluate chunk relevance contextually, lacking the deep reasoning needed to identify truly necessary information, which attention layers possess. Contextual info alone is insufficient.
>
> **7. Forward vs. Reverse Problems.**
>
> The difference isn't arithmetic ops, but implicit relationships (Sec 4.3, Fig 6). Forward problems use class-instance/hierarchical dependencies (concrete -> abstract). Reverse problems provide abstract values to find unknown instance values (abstract -> concrete). Most LLMs perform better on forward problems (App E), hypothesized due to training data bias towards constructive logic.
>
> We fine-tuned Qwen-2.5-500M with 1.3M generated problems (equal forward/reverse). The base model showed forward > reverse initially. Fine-tuning significantly boosted overall performance, with reverse slightly outperforming forward, confirming training data can address this. Detailed data in tables before and after fine-tuning is in **Sup** Part 3.

---

### Official Review · Reviewer_djUT · 2025-03-14

**Overall Recommendation:** 4

**Summary:**

This paper introduces GSM-∞, a synthetic long-context reasoning benchmark generated entirely by an automatic system with fine-grained control over complexity and information density. Specifically, it generates the benchmark by modifying operations like "+ -" in the computational graphs of existing benchmarks, which makes the generation method easily scalable. Extensive experiments on this benchmark further leads to several useful findings that are useful for further research in this direction.

## update after rebuttal
In the rebuttal, the authors answered my questions and provided an analysis that addresses my previous concerns. Therefore, i increased my score.

**Claims And Evidence:**

Yes

**Essential References Not Discussed:**

N/A

**Experimental Designs Or Analyses:**

Yes, I check the experiments in the corresponding sections.

**Methods And Evaluation Criteria:**

Yes.

**Other Comments Or Suggestions:**

1. Line 209, ” ->“
2. Line 431, the reference is not working

**Other Strengths And Weaknesses:**

Strengths:
1. The proposed benchmark generation method is directly applied on computational graphs of existing benchmarks, which makes it easily scalable in an automatic way.
2. The paper is written in a good flow, and experimental results have demonstrated the effectiveness of the proposed benchmark.
3. Several interesting findings such as the discussion on repeated sampling are presented.

Weaknesses:
1. There are no obvious weaknesses in this paper, except few typos that are listed below.

**Questions For Authors:**

Have the authors tried other reasoning relationships that can be obtained through the graph, like entailment?

**Relation To Broader Scientific Literature:**

The key contribution of the paper is related to a certain range of literature.

**Theoretical Claims:**

Yes, I check the correctness of such proofs and claims.

---

> ### Author Rebuttal · Authors · 2025-04-01
>
> **Response to other reasoning relationship question**
>
> We want to express our great appreciation to the reviewer for your positive feedback on GSM-Infinite. The question raised is also highly insightful and greatly appreciated. We want to clarify that the goal of GSM-Infinite is to model all the relationships that appear in GSM-8K with graph representation, an abstract way that is easier for manipulation and perturbation for scaling up in reasoning complexity and context length. GSM-Infinite model all explicit operations from plus, minus, multiply, and divide, and the implicit operations with hierarchical instance dependency. For example, “pigs” and “cats” are specific instances under the class “animal”, which GSM-Infinite can model well. (Is that the entailment relationship you are talking about?)
>
> On the other hand, GSM-Infinite by no means captures all the relationships possible through graph representations. We believe exploring these alternative complex relationships through a graph is also highly interesting and insightful. We will leave these explorations for further complex graph to natural language mappings to future works.

---

### Official Review · Reviewer_fqRq · 2025-03-15

**Overall Recommendation:** 3

**Summary:**

The paper introduces a new benchmark designed to evaluate the reasoning capabilities of Large Language Models (LLMs) in long and complex contexts.  The benchmark is inspired by the abstraction of GSM-8K problems as computational graphs and introduces a methodology to generate grade-school math problems with infinitely scalable difficulty and context length.  The authors evaluated existing LLMs, revealing a consistent sigmoid decline in reasoning performance as complexity increases, and an inefficiency in performance gains relative to inference computation scaling.

**Claims And Evidence:**

Mostly yes.

The first remaining concern is that the current dataset fit a rather simple pattern. It is unclear how models' performance would change if optimizing prompt or enabling certain tool use like code execution would change the performance and corresponding conclusion. Some failure mode analysis of a few LLMs is helpful to understand the possible flaws of the dataset.

The second concern is that it is unclear how the proposed dataset reflects real use case of reasoning models.

**Essential References Not Discussed:**

NA

**Experimental Designs Or Analyses:**

Please check above for details.

**Methods And Evaluation Criteria:**

No concerns.

**Other Comments Or Suggestions:**

NA

**Other Strengths And Weaknesses:**

NA

**Questions For Authors:**

Please check above for details.

**Relation To Broader Scientific Literature:**

NA

**Theoretical Claims:**

The authors claimed infite length and mentioned reduced probability of multiple solutions but some formal analysis on the cost of generating a valid question w.r.t. length and the actual probability of such cases should be provided.

---

> ### Author Rebuttal · Authors · 2025-04-01
>
> **Concern GSM-Infinite not practical**
>
> We appreciate the reviewer’s concern about GSM-Infinite’s practicality. While GSM-8K has been a standard for evaluating LLM math reasoning, its difficulty has become saturated (Figure 5(a), Page 4), which is why newer models have moved on. GSM-Infinite builds on GSM-8K by abstracting its core operations into graph representations and using graph perturbation to scale both reasoning complexity and context length. Our pipeline is entirely LLM- and human-free, generated independently of GSM-8K (Section 3, Pages 3–5). By scaling up the op=200 and above, we saturate SOTA LLMs and capture their full spectrum of reasoning ability for study.
>
> On the other hand, recent LLMs often report metrics from MATH or AIME datasets. Although these datasets are of strong reasoning complexity, solving test examples convincingly requires a lot of math-related theorem knowledge in LLMs’ memory. GSM-Infinite offers an alternative reasoning benchmark that is drastically less reliant on LLMs’ memory, while evaluating all LLMs based solely on their reasoning ability of traversing a complex computational graph problem with fine-grained difficulty control for the study.
>
> Because of space limitations, we hold five examples for op=5 for both Llama-3.1-8B-Instruct and Qwen-2.5-7B-Instruct in the following link: https://docs.google.com/document/d/1WP3ygB67yNUS-iliSYYjFRY4Ls81yIVcRVxpbTVTZ0o/edit?usp=sharing.
>
> We select a smaller operation setting with a smaller model to keep the complexity of the problem lower for the practicality of human inspection. Below is a summary of error causes and the corresponding index of the example.
> | Error Type                                                             | Index of the examples     |
> |------------------------------------------------------------------------|---------------------------|
> | Confuses two similar but distinctive concepts in the question          | (1), (8), (10)            |
> | Hallucinate a concept in the question, which the question does not mention | (2), (4), (9)             |
> | Get distracted by unnecessary variables in the question                | (3)                       |
> | Misinterpret the relationship mentioned in the question                | (5), (6), (7)             |
>
> **Generation Throughput**
>
> During generation, we strictly constrain the number range in the solution path and disallow negative or floating-point numbers in arithmetic operations. These constraints make generating the Hard subset of GSM-Infinite significantly slower than Symbolic and Medium, as Hard requires heavy use of aggregation, implicit multiplication and addition when op is large—which quickly inflate numbers and reduce flexibility compared to Symbolic, where operations can be mixed more freely. Despite this, our system achieves far superior throughput on the Hard subset compared to any LLM- or human-in-the-loop approach. Using 16 threads on an M3 Pro, we reach 1.1M tokens/s for examples averaging 1k tokens (op=50), a speed that's hard to match with LLM-based generation.
>
> | op  | Throughput (Valid Examples/s) |
> |-----|-------------------------------|
> | 10  | 5510.88                       |
> | 20  | 6747.20                       |
> | 30  | 1919.68                       |
> | 40  | 2228.00                          |
> | 50  | 1149.92                       |
>
> When going up the op number, the throughput will go down because the chance that the operation will avoid the strict rejection sampling criteria is decreasing, causing some wasted computation. Just for reference, the much more flexible Symbolic subset can generate problems with 30k examples/s with 16 threads on CPU, since we can have more precise control of every operation in the Symbolic benchmark. We think the need for strong rejection sampling criteria is an implementation challenge, and more careful and more efficient graph control implementation is greatly possible and likely to lead to further improvement in throughput.
>
> |Context Length|Throughput (Valid Example/s)|
> |---|---|
> |Zero noise|6747.20|
> |8K|3896.48|
> |16K|3932.96|
> |32K|3898.08|
> |64K|3917.60|
> |128K|3954.18|
>
> The above shows throughput for op=20 for increasing context length. The sudden drop in throughput from zero noise to 8k is because when injecting noise to a specific context length requirement, we have to first perform a tokenization step of the zero-noise example and calculate how much noise statement we need. The one extra tokenization causes a slowdown in generation throughput. On the other hand, since we impose far less strict checking on the noise statements, generating noise is highly efficient and parallelizable across multiple threads. We show our generator’s strong context length flexibility.

---

> > ### Comment · Reviewer_fqRq · 2025-04-05
> >
> > The reviewer appreciates the rebuttal. But the practical value of the proposed benchmark is still concerning for the reviewer.
> >
> > Although the authors argue that GSM-8k  has been widely used nowadays, the reviewer still finds the proposed dataset distant from the real use case of long reasoning process, where much more broad knowledge or skill is needed and it is intellectual challenging to reach certain intermediate steps. But this is not the case with the current proposed evaluation. This is what the reviewer summarized as far from real use cases.
> >
> > The reviewer keeps the score because this current simple pattern could be still useful to probe long-context ability.

---

> > > ### Author Response · Authors · 2025-04-05
> > >
> > > We appreciate the reviewer's follow-up comments.
> > >
> > > We want to kindly rebut that synthetic long-context benchmarks are highly popular today (Needle-in-a-haystack, RULER, etc). These tasks evaluate the fundamental ability of the LLM needed for more realistic downstream tasks. Needle-in-a-haystack, for example, evaluates the fundamental ability of LLM to retrieve a certain chunk in the context given the query. Failure in doing so suggests the LLM's fundamental inability to tackle tasks that are more realistic and intellectually challenging. **The synthetic tasks have helped many recent LLM developments to better benchmark their models' long-context abilities and have become universally accepted by the frontier LLMs for demonstrating their long-context ability** (e.g., Google Gemini 1.5 and later models)
> > >
> > > For GSM-Infinite, we first identify that although models are getting a perfect score on Needle-in-a-haystack, RULER, LongBench, etc, **it is far from claiming that the models are fundamentally strong in long-context tasks**. As we have shown (Section 2, page 3) that across most popular long-context benchmarks, RAGs, which are much cheaper in inference than LLMs, has performance on-par, or even better than LLMs. We need a harder benchmark that evaluates the true value of LLMs over cheap RAG methods to further facilitate the long-context LLM development. **We also have shown that constructing a harder benchmark isn't straightforward**, as it is non-trivial to convert short-context hard problems to long-context ones by injecting random noise, as that would even help RAGs beat long-context LLMs (Figure 7 (d), page 8).
> > >
> > > Following the prior synthetic long-context benchmarks that are heavily popular, we also want to design benchmarks that evaluate the fundamental reasoning capability of these LLMs. **We intentionally ensure that solving the problem correctly doesn't require domain-specific knowledge or high-level tool-using.** Therefore, we can know for certain that making a mistake on the benchmark is caused by the fundamental information extraction or understanding rather than unfamiliarity with the knowledge during pretraining. Grade-school math can be abstracted into fundamental operations and relationships (confirmed by previous work [1]). We take these operations and develop a scalable generator that can map all relationships that appear in GSM-8K to the generated problems that are LLM and human understandable. Since the operation is well-defined, we can evaluate LLM with greater precision on their behaviors. Also, the generator is crucial for developing long-context filler text that is deeply relevant to the main problem, such that RAGs are unable to filter (Figure 4(c), page 4).
> > >
> > > Admittedly, we fully agree that the benchmark needs broad knowledge and challenging skills. **However, we want to point out that naively collecting this real-world data is impractical at a large scale, as we have discussed and cited SWE-bench and DafnyBench.** They required a huge amount of human engineering for cleaning and deduplication while still limited by diversity and quantity. We argue that synthetic generation offers another possible path towards these benchmarks for its high controllability and flexibility in generation. GSM-Infinite offers a solid step towards that goal.
> > >
> > > [1] Ye, T., Xu, Z., Li, Y., & Allen-Zhu, Z. (2024, July). Physics of language models: Part 2.1, grade-school math and the hidden reasoning process. In The Thirteenth International Conference on Learning Representations.

---

### Official Review · Reviewer_6KRZ · 2025-03-25

**Overall Recommendation:** 3

**Summary:**

This paper points out issues in existing long-context reasoning benchmarks and addresses the issues by designing new benchmark called GSM-Infinite.
The main method they use is to construct computation graphs for problems in GSM8K and generate question-answer pairs with user-definable question difficulty measured by reasoning steps.
The main findings is that LLMs' performance decays with reasoning complexity, addition of noise. They struggle more on backward thinking tasks and benefit from more inference steps.
For model performance, they find that reasoning models do better than non-reasoning LLMs.

**Claims And Evidence:**

Claims they made:
1) LLM Performance Degradation Can be Modeled Using Sigmoid Function
2) Reverse Problems are Harder to Solve for LLMs
3) LLMs' performance degrade as the context grows long and as noise is added.
All of them are supported by experimental results.

**Essential References Not Discussed:**

This paper does not provide discussion on multi-step reasoning literature.
There are several classical benchmarks which should be included in the paper:

[1] Sinha, Koustuv, et al. "CLUTRR: A diagnostic benchmark for inductive reasoning from text." arXiv preprint arXiv:1908.06177 (2019).
[2] Yang, Zhilin, et al. "HotpotQA: A dataset for diverse, explainable multi-hop question answering." arXiv preprint arXiv:1809.09600 (2018).
[3] Trivedi, Harsh, et al. "♫ MuSiQue: Multihop Questions via Single-hop Question Composition." Transactions of the Association for Computational Linguistics 10 (2022): 539-554.

**Experimental Designs Or Analyses:**

See Questions.

**Methods And Evaluation Criteria:**

Their evaluation method for different LLMs with different context windows make sense.

**Other Comments Or Suggestions:**

Some of the references are repeated or having inconsistent formatting.
e.g.

Kamradt, G. Needle in a haystack - pressure testing llms, 2023. URL https://github.com/gkamradt/LLMTestNeedleInAHaystack/tree/main.
is repeated.

Chieh. et al. 2024a and 2024b are the same.

**Other Strengths And Weaknesses:**

The authors did a good job explaining details e.g. how to build computation graph for reasoning problems including how to handle different operations and how to add noise.
However, I feel that it lacks a good explanation on how the whole pipeline works. See Questions.

**Questions For Authors:**

It is possible that I missed the relevant part, but I am confused about the RAG setting: what is treated as a single "article"? Is it a single graph?
Why is it needed that that LLMs have 8k context minimum? Are you putting all the available graphs and generate questions at one shot?
What is the intuition behind "reverse problems are harder to solve"? After the problems are generated properly using the computation graph, shouldn't it be the same for the models to solve dividing and minus math problems as solving the addition and multiply types?

**Relation To Broader Scientific Literature:**

I think this paper is related to the long-context LLM reading comprehension literature and also multi-hop reasoning benchmarks.

**Theoretical Claims:**

There is no theoretical claims in this paper.

---

> ### Author Rebuttal · Authors · 2025-04-01
>
> **Three missing citations**
>
> Greatly appreciate the comments. We acknowledge that more discussion of previous multi-hop commonsense reasoning benchmarks **will be added in the paper later version**. But these three papers address different problems from ours.
>
> Firstly, HotpotQA [2] and MusiQue [3] are on short-context 2 to 4-hop commonsense reasoning. Also, huge human labor is required to collect, limiting scaling up on quantity or the reasoning complexity. Also, these two are converted to long-context problems incorporated in RULER ([2]) and LOFT ([2] and [3]), which are cited and heavily studied (Section 2, page 3). The added noise in two can be effectively filtered by RAG, inapt for LC-LLM evaluation.
>
> Secondly, CLUTRR [1], though aired before LLMs, similarly generates problems of family relationship deduction from graphs but with 3 big distinctions. (1) Despite similarity in methods, CLUTRR is heavily tailored to family relation deduction, especially specialized to kinship graphs, severely limited by scope.
> (2) Human annotation is essential to making CLUTRR, while GSM-Infinite relies completely on software, no LLM nor humans. No guarantee of the absolute correctness of the labels provided by the generator is a crucial drawback, as humans make mistakes. Specifically, we found that a minor part of the dataset is mislabeled. One example below.
>
> *Label: sister
> Question: Given the story, please provide the relationship between two selected people. Story: [James] bought a new dress for his daughter [Lynn]. [Theodore] and his son [Wesley] went to look at cars. [Theodore] ended up buying the Mustang. [James] took his grandson [Theodore] to the baseball game. Question: What is the relationship between 'Wesley' and 'Lynn'?*
>
> The corrected label is Wesley is the grand nephew of Lynn.
>
> (3) Having either LLM or Human in the loop also leads to huge difficulty in scaling up the reasoning complexity. Below, we evaluate the 2to10 test set given by CLUTRR with various LLMs.
> | Models   | Accuracy |
> |---|---|
> | Llama 3.1 8B Instruct | 0.64     |
> | GPT-4o  | 0.7      |
>
> Lack of scaling prevents the evaluation of the full spectrum of behaviors. GSM-Infinite generates op=200 problems (and more if needed) to capture the full spectrum of SOTA models' reasoning performance.
>
> **RAG clarification**
>
> We follow the convention of context-level RAG method [1]. We solely use the input context as the data store (no external source). We segment the input text into chunks (roughly a sentence) using NLTK package. The retriever first rates each chunk based on its relevance to the query, and then the top-k scoring chunks are selected as the input to the decoder model. k is determined by the 2k budget from our experiments.
>
> [1] Lewis, P. Retrieval-augmented generation for knowledge-intensive nlp tasks. (2020)
>
> **Dataset generation with Graph Clarification**
>
> During generation, no human/LLM is in the loop. “One-shot” generation is irrelevant to our settings. Each problem, regardless of context length, is generated based on one computational graph (not multiple graphs). Natural language problems directly mapped from the graph using our developed generator software. For a detailed explanation and reference, please look at (Figure 6, page 6) and (Section 3.1, 3.2, page 3-5).
>
> **Forward Reverse Observation**
>
> The difference between forward and reverse problems isn’t the arithmetic operations. Referring to (Section 4.3, page 6), the explicit plus, minus, multiply, and divide operations appear in both forward and reverse problems. **The difference is the implicit operation/relationships contained.** The implicit addition and multiplication are displaying class-instance and hierarchical instance dependency, respectively. Problems in forward logic only contain these two implicit operations - you can notice that in the computation graph (Figure 6, page 6), the more concrete and detailed instances are first computed, and then, using them collectively, more general and summarizing variables are computed. However, reverse problems contain the implicit minus and division operations. Given the value for the abstract variables, the LLM needs to compute the unknown instance variables. Here is a simplified example.
> Forward Problem - #pigs in Zoo is 2. #sheeps in Zoo is 4. Assume the animals not mentioned are of 0 quantity. What is the #total animals in Zoo?
> Solution: We go from #pigs in Zoo (instance) -> #sheeps in Zoo (instance) -> total animals (abstract).
> Reverse Problem - #pigs in Zoo is 2. #Sheep in Zoo is non-zero. #total animals in Zoo is 6. #sheep in Zoo?
> Solution: We go from total animals (abstract) -> #pigs in Zoo (instance) -> #sheeps in Zoo (instance).
> We show a detailed study in (Appendix E, page 18). We hypothesize that models' usual strength in forward stems from that humans naturally write in the constructive logic ordering, or forward, more, leading to naturally less training data for reverse logics.

---

> > ### Comment · Reviewer_6KRZ · 2025-04-08
> >
> > Thanks for the replay and clarification.
> >
> > My concerns are mostly addressed and I will increase my score.

---

> > > ### Author Response · Authors · 2025-04-09
> > >
> > > Thank you so much for your positive review on GSM-Infinite!

---

### Decision · Program_Chairs · 2025-05-01

**Decision:**

Accept (poster)

**Comment:**

**Summary**: This paper designs a way to controllably generate GSM8K-type problems with varying context length / difficulty / reasoning. The paper then analyzes 18 SoTA LLMs on the generated problems and reports how their performance varies with varying problem complexity, varying types of distractors injected into the problem, varying model sizes/types and also varying inference-time compute (Best-of-N sampling, specifically).

The controllable and scalable benchmark here would be highly valuable for analyses; despite the GSM8K-ness of the benchmark, it makes good progress on top of existing benchmarks; also, the experiments are comprehensive and the reviewers did not find any major flaws.

**Details**:
Reviewers appreciated the value in how the work here complements other reasoning benchmarks (`GB4k`), isolates long-context reasoning from long-context retrieval (`6Z5g`), and makes data generation scalable (`djUT`). The paper describes its process and findings well (`6KRZ`, `djUT`); the experiments are designed well (`GB4k`); they demonstrate the claims (`djUT`), are comprehensive (`6Z5g `) and the findings are interesting (`djUT`). They did not find any major weaknesses (`fqRq`, `GB4k`) including Reviewer `GB4k` who read the appendix carefully.  I would add that the experimental observations themselves are as one would expect a priori; but there is significant value in establishing them rigorously and comprehensively.

The one concern that stood out as important to me was that the data generated here may be too simple (`fqRq`) or too tailored to GSM8K (`GB4k`) affecting the significance of the dataset. However, I believe that in the context of what exists in literature, this is relatively more complex and is valuable progress. The authors contextualize their work very well in the introduction: existing long-context datasets can either be solved via retrieval as they have easy-to-identify distractors or are not scalable to infinite complexity/infinite data.

Suggestions: Reviewer `6Z5g` had a valuable suggestion about varying the computational graph structures which the authors have responded to. Reviewers `6Z5g`, `GB4k` both requested a qualitative analysis of model errors which the authors responded to. I encourage the authors to present a condensed version of this in the paper. I agree with `6Z5g` regarding improving the description of the computation graph construction. This part was hard to follow. Concretely, it would help to (a) provide more varying examples of two-entity and three-entity variables and (b) be more careful about discussing the intuition (e.g., the argument in the first para of 4.1 Reverse Problems was hard to follow).

Minor note: I would use the word "distractor" or "filler" rather than "noise". Noise to me indicates corrupting the essential values which is different from what is being studied here.